# Continuous Hierarchical Representations with Poincaré Variational Auto-Encoders

**Emile Mathieu**[†]
emile.mathieu@stats.ox.ac.uk

**Charline Le Lan**[†]
charline.lelan@stats.ox.ac.uk

**Chris J. Maddison**[†*]
cmaddis@stats.ox.ac.uk

**Ryota Tomioka**[‡]
ryoto@microsoft.com

**Yee Whye Teh**[†*]
y.w.teh@stats.ox.ac.uk

† Department of Statistics, University of Oxford, United Kingdom
∗ DeepMind, London, United Kingdom
‡ Microsoft Research, Cambridge, United Kingdom

## Abstract

The variational auto-encoder (VAE) is a popular method for learning a generative model and embeddings of the data. Many real datasets are hierarchically structured. However, traditional VAEs map data in a Euclidean latent space which cannot efficiently embed tree-like structures. Hyperbolic spaces with negative curvature can. We therefore endow VAEs with a Poincaré ball model of hyperbolic geometry as a latent space and rigorously derive the necessary methods to work with two main Gaussian generalisations on that space. We empirically show better generalisation to unseen data than the Euclidean counterpart, and can qualitatively and quantitatively better recover hierarchical structures.

## 1 Introduction

Learning useful representations from unlabelled raw sensory observations, which are often high-dimensional, is a problem of significant importance in machine learning. Variational auto-encoders (VAEs) (Kingma and Welling, 2014; Rezende et al., 2014) are a popular approach to this: they are probabilistic generative models composed of an *encoder* stochastically embedding observations in a low dimensional latent space $\mathcal{Z}$, and a *decoder* generating observations $x \in \mathcal{X}$ from encodings $z \in \mathcal{Z}$. After training, the encodings constitute a low-dimensional representation of the original raw observations, which can be used as features for a downstream task (e.g. Huang and LeCun, 2006; Coates et al., 2011) or be interpretable for their own sake. VAEs are therefore of interest for representation learning (Bengio et al., 2013), a field which aims to learn *good representations*, e.g. interpretable representations, ones yielding better generalisation, or ones useful for downstream tasks.

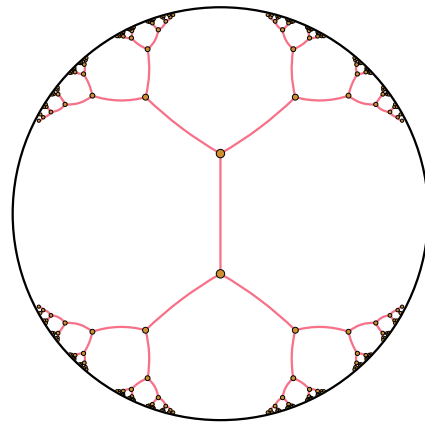

Figure 1: A regular tree isometrically embedded in the Poincaré disc. Red curves are same length *geodesics*, i.e. "straight lines".

It can be argued that in many domains data should be represented hierarchically. For example, in cognitive science, it is widely accepted that human beings use a hierarchy to organise object categories (e.g. Roy et al., 2006; Collins and Quillian, 1969; Keil, 1979). In biology, the theory of evolution (Darwin, 1859) implies that features of living organisms are related in a hierarchical manner given by the evolutionary tree. Explicitly incorporating hierarchical structure in probabilistic models has unsurprisingly been a long-running research topic (e.g. Duda et al., 2000; Heller and Ghahramani, 2005).

Earlier work in this direction tended to use trees as data structures to represent hierarchies. Recently, hyperbolic spaces have been proposed as an alternative continuous approach to learn hierarchical representations from textual and graph-structured data (Nickel and Kiela, 2017; Tifrea et al., 2019). Hyperbolic spaces can be thought of as continuous versions of trees, and vice versa, as illustrated in Figure 1. Trees can be embedded with arbitrarily low error into the Poincaré disc model of hyperbolic geometry (Sarkar, 2012). The exponential growth of the Poincaré surface area with respect to its radius is analogous to the exponential growth of the number of leaves in a tree with respect to its depth. Further, these spaces are smooth, enabling the use of deep learning approaches which rely on differentiability.

We show that replacing VAEs latent space components, which traditionally assume a Euclidean metric over the latent space, by their hyperbolic generalisation helps to represent and discover hierarchies. Our goals are twofold: (a) learn a latent representation that is interpretable in terms of hierarchical relationships among the observations, (b) learn a more efficient representation which generalises better to unseen data that is hierarchically structured. Our main contributions are as follows:

1. We propose efficient and reparametrisable sampling schemes, and calculate the probability density functions, for two canonical Gaussian generalisations defined on the Poincaré ball, namely the maximum-entropy and wrapped normal distributions. These are the ingredients required to train our VAEs.

2. We introduce a decoder architecture that explicitly takes into account the hyperbolic geometry, which we empirically show to be crucial.

3. We empirically demonstrate that endowing a VAE with a Poincaré ball latent space can be beneficial in terms of model generalisation and can yield more interpretable representations.

Our work fits well with a surge of interest in combining hyperbolic geometry and VAEs. Of these, it relates most strongly to the concurrent works of Ovinnikov (2018); Grattarola et al. (2019); Nagano et al. (2019). In contrast to these approaches, we introduce a decoder that takes into account the geometry of the hyperbolic latent space. Along with the *wrapped normal* generalisation used in the latter two articles, we give a thorough treatment of the *maximum entropy normal* generalisation and a rigorous analysis of the difference between the two. Additionally, we train our model by maximising a lower bound on the marginal likelihood, as opposed to Ovinnikov (2018); Grattarola et al. (2019) which consider a Wasserstein and an adversarial auto-encoder setting, respectively. We discuss these works in more detail in Section 4.

## 2 The Poincaré Ball model of hyperbolic geometry

### 2.1 Review of Riemannian geometry

Throughout the paper we denote the Euclidean norm and inner product by $\|\cdot\|$ and $\langle \cdot, \cdot \rangle$ respectively. A real, smooth *manifold* $\mathcal{M}$ is a set of points $z$, which is "locally similar" to a linear space. For every point $z$ of the manifold $\mathcal{M}$ is attached a real vector space of the same dimensionality as $\mathcal{M}$ called the *tangent space* $\mathcal{T}_z\mathcal{M}$. Intuitively, it contains all the possible directions in which one can tangentially pass through $z$. For each point $z$ of the manifold, the *metric tensor* $\mathfrak{g}(z)$ defines an inner product on the associated tangent space : $\mathfrak{g}(z) = \langle \cdot, \cdot \rangle_z : \mathcal{T}_z\mathcal{M} \times \mathcal{T}_z\mathcal{M} \to \mathbb{R}$. The *matrix representation of the Riemannian metric* $G(z)$, is defined such that $\forall u, v \in \mathcal{T}_z\mathcal{M} \times \mathcal{T}_z\mathcal{M}$, $\langle u, v \rangle_z = \mathfrak{g}(z)(u, v) = u^T G(z) v$. A *Riemannian manifold* is then defined as a tuple $(\mathcal{M}, \mathfrak{g})$ (Petersen, 2006). The metric tensor gives a *local* notion of angle, length of curves, surface area and volume, from which *global* quantities can be derived by integrating local contributions. A norm is induced by the inner product on $\mathcal{T}_z\mathcal{M}$: $\|\cdot\|_z = \sqrt{\langle \cdot, \cdot \rangle_z}$. An infinitesimal volume element is induced on each tangent space $\mathcal{T}_z\mathcal{M}$, and thus a measure $d\mathcal{M}(z) = \sqrt{|G(z)|}dz$ on the manifold, with $dz$ being the Lebesgue measure.

The length of a curve $\gamma : t \mapsto \gamma(t) \in \mathcal{M}$ is given by $L(\gamma) = \int_0^1 \|\gamma'(t)\|_{\gamma(t)}^{1/2} dt$. The concept of straight lines can then be generalised to *geodesics*, which are constant speed curves giving the shortest path between pairs of points $\boldsymbol{z}, \boldsymbol{y}$ of the manifold: $\gamma^* = \arg\min L(\gamma)$ with $\gamma(0) = \boldsymbol{z}$, $\gamma(1) = \boldsymbol{y}$ and $\|\gamma'(t)\|_{\gamma(t)} = 1$. A *global* distance is thus induced on $\mathcal{M}$ given by $d_{\mathcal{M}}(\boldsymbol{z}, \boldsymbol{y}) = \inf L(\gamma)$. Endowing $\mathcal{M}$ with that distance consequently defines a metric space $(\mathcal{M}, d_{\mathcal{M}})$. The concept of moving along a "straight" curve with constant velocity is given by the *exponential map*. In particular, there is a unique unit speed *geodesic* $\gamma$ satisfying $\gamma(0) = \boldsymbol{z}$ with initial tangent vector $\gamma'(0) = \boldsymbol{v}$. The corresponding exponential map is then defined by $\exp_{\boldsymbol{z}}(\boldsymbol{v}) = \gamma(1)$, as illustrated on Figure 2. The *logarithm map* is the inverse $\log_{\boldsymbol{z}} = \exp_{\boldsymbol{z}}^{-1} : \mathcal{M} \to \mathcal{T}_{\boldsymbol{z}}\mathcal{M}$. For geodesically complete manifolds, such as the Poincaré ball, $\exp_{\boldsymbol{z}}$ is well-defined on the full tangent space $\mathcal{T}_{\boldsymbol{z}}\mathcal{M}$ for all $z \in \mathcal{M}$.

## 2.2 The Poincaré ball model of hyperbolic geometry

A $d$-dimensional hyperbolic space, denoted $\mathbb{H}^d$, is a complete, simply connected, $d$-dimensional Riemannian manifold with constant negative curvature $c$. In contrast with the Euclidean space $\mathbb{R}^d$, $\mathbb{H}^d$ can be constructed using various isomorphic models (none of which is prevalent), including the hyperboloid model, the Beltrami-Klein model, the Poincaré half-plane model and the Poincaré ball $\mathcal{B}_c^d$ (Beltrami, 1868). The Poincaré ball model is formally defined as the Riemannian manifold $\mathbb{B}_c^d = (\mathcal{B}_c^d, \mathfrak{g}_p^c)$, where $\mathcal{B}_c^d$ is the open ball of radius $1/\sqrt{c}$, and $\mathfrak{g}_p^c$ its *metric tensor*, which along with its induced distance are given by

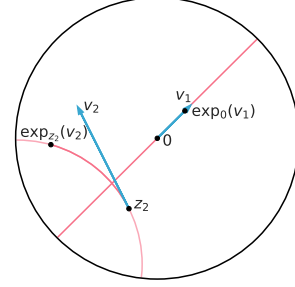

Figure 2: Geodesics and exponential maps in the Poincaré disc.

$$\mathfrak{g}_p^c(\boldsymbol{z}) = (\lambda_{\boldsymbol{z}}^c)^2 \, \mathfrak{g}_e(\boldsymbol{z}), \quad d_p^c(\boldsymbol{z}, \boldsymbol{y}) = \frac{1}{\sqrt{c}} \cosh^{-1}\left(1 + 2c\frac{\|\boldsymbol{z} - \boldsymbol{y}\|^2}{(1 - c\|\boldsymbol{z}\|^2)(1 - c\|\boldsymbol{y}\|^2)}\right),$$

where $\lambda_{\boldsymbol{z}}^c = \frac{2}{1 - c\|\boldsymbol{z}\|^2}$ and $\mathfrak{g}_e$ denotes the Euclidean metric tensor, i.e. the usual dot product. The *Möbius addition* (Ungar, 2008) of $\boldsymbol{z}$ and $\boldsymbol{y}$ in $\mathbb{B}_c^d$ is defined as

$$\boldsymbol{z} \oplus_c \boldsymbol{y} = \frac{(1 + 2c \langle \boldsymbol{z}, \boldsymbol{y} \rangle + c\|\boldsymbol{y}\|^2)\boldsymbol{z} + (1 - c\|\boldsymbol{z}\|^2)\boldsymbol{y}}{1 + 2c \langle \boldsymbol{z}, \boldsymbol{y} \rangle + c^2\|\boldsymbol{z}\|^2\|\boldsymbol{y}\|^2}.$$

One recovers the Euclidean addition of two vectors in $\mathbb{R}^d$ as $c \to 0$. Building on that framework, Ganea et al. (2018) derived closed-form formulations for the *exponential map* (illustrated in Figure 2)

$$\exp_{\boldsymbol{z}}^c(\boldsymbol{v}) = \boldsymbol{z} \oplus_c \left(\tanh\left(\sqrt{c}\frac{\lambda_{\boldsymbol{z}}^c\|\boldsymbol{v}\|}{2}\right) \frac{\boldsymbol{v}}{\sqrt{c}\|\boldsymbol{v}\|}\right)$$

and its inverse, the *logarithm map*

$$\log_{\boldsymbol{z}}^c(\boldsymbol{y}) = \frac{2}{\sqrt{c}\lambda_{\boldsymbol{z}}^c} \tanh^{-1}\left(\sqrt{c}\| - \boldsymbol{z} \oplus_c \boldsymbol{y}\|\right) \frac{-\boldsymbol{z} \oplus_c \boldsymbol{y}}{\| - \boldsymbol{z} \oplus_c \boldsymbol{y}\|}.$$

# 3 The Poincaré VAE

We consider the problem of mapping an empirical distribution of observations to a lower dimensional Poincaré ball $\mathbb{B}_c^d$, as well as learning a map from this latent space $\mathcal{Z} = \mathbb{B}_c^d$ to the observation space $\mathcal{X}$. Building on the VAE framework, this *Poincaré*-VAE model, or $\mathcal{P}^c$-VAE for short, differs by the choice of prior and posterior distributions being defined on $\mathbb{B}_c^d$, and by the encoder $g_\phi$ and decoder $f_\theta$ maps which take into account the latent space geometry. Their parameters $\{\theta, \phi\}$ are learned by maximising the *evidence lower bound* (ELBO). Our model can be seen as a generalisation of a classical Euclidean VAE (Kingma and Welling, 2014; Rezende et al., 2014) that we denote by $\mathcal{N}$-VAE, i.e. $\mathcal{P}^c$-VAE $\xrightarrow[c \to 0]{} \mathcal{N}$-VAE.

## 3.1 Prior and variational posterior distributions

In order to parametrise distributions on the Poincaré ball, we consider two canonical generalisations of normal distributions on that space. A more detailed review of Gaussian generalisations on manifolds can be found in Appendix B.1.

**Riemannian normal** One generalisation is the distribution maximising entropy given an expectation and variance (Said et al., 2014; Pennec, 2006; Hauberg, 2018), often called the *Riemannian normal* distribution, which has a density w.r.t. the metric induced measure $d\mathcal{M}$ given by

$$\mathcal{N}^{\mathrm{R}}_{\mathbb{B}^d_c}(\boldsymbol{z}|\boldsymbol{\mu},\sigma^2) = \frac{d\nu^{\mathrm{R}}(\boldsymbol{z}|\boldsymbol{\mu},\sigma^2)}{d\mathcal{M}(\boldsymbol{z})} = \frac{1}{Z^{\mathrm{R}}}\exp\left(-\frac{d^c_p(\boldsymbol{\mu},\boldsymbol{z})^2}{2\sigma^2}\right), \quad (1)$$

where $\sigma > 0$ is a dispersion parameter, $\boldsymbol{\mu} \in \mathbb{B}^d_c$ is the Fréchet mean , and $Z^{\mathrm{R}}$ is the normalising constant derived in Appendix B.4.3.

**Wrapped normal** An alternative is to consider the push-forward measure obtained by mapping a normal distribution along the *exponential map* $\exp_{\boldsymbol{\mu}}$. That probability measure is often referred to as the *wrapped normal* distribution, and has been used in auto-encoder frameworks with other manifolds (Grattarola et al., 2019; Nagano et al., 2019; Falorsi et al., 2018). Samples $\boldsymbol{z} \in \mathbb{B}^d_c$ are obtained as $\boldsymbol{z} = \exp^c_{\boldsymbol{\mu}}(\boldsymbol{v}/\lambda^c_{\boldsymbol{\mu}})$ with $\boldsymbol{v} \sim \mathcal{N}(\cdot|\boldsymbol{0},\Sigma)$ and its density is given by (details given in Appendix B.3)

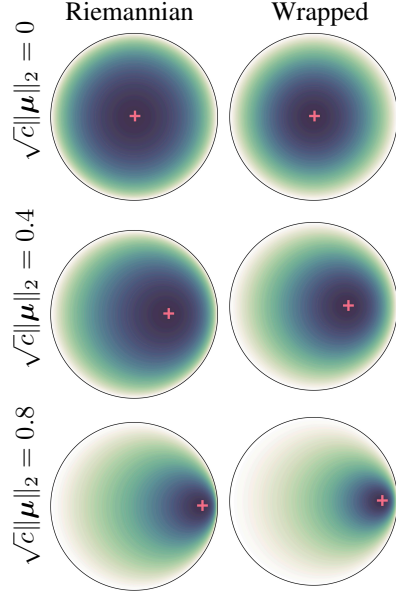

$$\mathcal{N}^{\mathrm{W}}_{\mathbb{B}^d_c}(\boldsymbol{z}|\boldsymbol{\mu},\Sigma) = \frac{d\nu^{\mathrm{W}}(\boldsymbol{z}|\boldsymbol{\mu},\Sigma)}{d\mathcal{M}(\boldsymbol{z})} \quad (2)$$

$$= \mathcal{N}\left(\lambda^c_{\boldsymbol{\mu}}\log_{\boldsymbol{\mu}}(\boldsymbol{z})|\boldsymbol{0},\Sigma\right)\left(\frac{\sqrt{c}\,d^c_p(\boldsymbol{\mu},\boldsymbol{z})}{\sinh(\sqrt{c}\,d^c_p(\boldsymbol{\mu},\boldsymbol{z}))}\right)^{d-1}.$$

The (usual) normal distribution is recovered for both generalisations as $c \to 0$. We discuss the benefits and drawbacks of those two distributions in Appendix B.1. We refer to both as *hyperbolic normal* distributions with pdf $\mathcal{N}_{\mathbb{B}^d_c}(\boldsymbol{z}|\boldsymbol{\mu},\sigma^2)$. Figure 8 shows several probability densities for both distributions.

The prior distribution defined on $\mathcal{Z}$ is chosen to be a hyperbolic normal distribution with mean zero, $p(\boldsymbol{z}) = \mathcal{N}_{\mathbb{B}^d_c}(\cdot|\boldsymbol{0},\sigma^2_0)$, and the variational family is chosen to be parametrised as $\mathcal{Q} = \{\mathcal{N}_{\mathbb{B}^d_c}(\cdot|\boldsymbol{\mu},\sigma^2)\mid\boldsymbol{\mu}\in\mathbb{B}^d_c,\sigma\in\mathbb{R}^+_*\}$.

Figure 3: Hyperbolic normal probability density for different Fréchet mean, same standard deviation and $c = 10$. The *Riemannian* hyperbolic radius has a slightly larger mode.

## 3.2 Encoder and decoder

We make use of two neural networks, a *decoder* $f_{\boldsymbol{\theta}}$ and an *encoder* $g_{\boldsymbol{\phi}}$, to parametrise the likelihood $p(\cdot|f_{\boldsymbol{\theta}}(\boldsymbol{z}))$ and the variational posterior $q(\cdot|g_{\boldsymbol{\phi}}(\boldsymbol{x}))$ respectively. The input of $f_{\boldsymbol{\theta}}$ and the output of $g_{\boldsymbol{\phi}}$ need to respect the hyperbolic geometry of $\mathcal{Z}$. In the following we describe appropriate choices for the first layer of the decoder and the last layer of the encoder.

**Decoder** In the Euclidean case, an affine transformation can be written in the form $f_{\boldsymbol{a},\boldsymbol{p}}(\boldsymbol{z}) = \langle\boldsymbol{a},\boldsymbol{z}-\boldsymbol{p}\rangle$, with orientation and offset parameters $\boldsymbol{a},\boldsymbol{p} \in \mathbb{R}^d$. This can be rewritten in the form

$f_{\boldsymbol{a},\boldsymbol{p}}(\boldsymbol{z}) = \mathrm{sign}(\langle\boldsymbol{a},\boldsymbol{z}-\boldsymbol{p}\rangle)\,\|\boldsymbol{a}\|\,d_E(\boldsymbol{z},H^c_{\boldsymbol{a},\boldsymbol{p}})$

where $H_{\boldsymbol{a},\boldsymbol{p}} = \{\boldsymbol{z} \in \mathbb{R}^p \mid \langle\boldsymbol{a},\boldsymbol{z}-\boldsymbol{p}\rangle = 0\} = \boldsymbol{p}+\{\boldsymbol{a}\}^{\perp}$ is the decision hyperplane. The third term is the distance between $\boldsymbol{z}$ and the decision hyperplane $H^c_{\boldsymbol{a},\boldsymbol{p}}$ and the first term refers to the side of $H^c_{\boldsymbol{a},\boldsymbol{p}}$ where $\boldsymbol{z}$ lies. Ganea et al. (2018) analogously introduced an operator $f^c_{\boldsymbol{a},\boldsymbol{p}} : \mathbb{B}^d_c \to \mathbb{R}^p$ on the Poincaré ball,

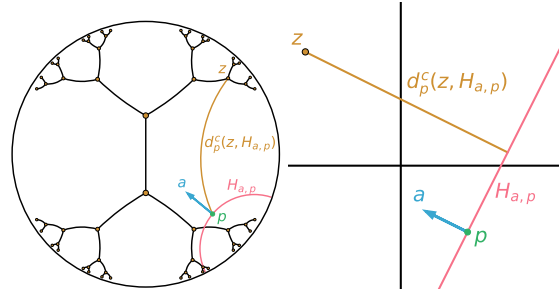

Figure 4: Illustration of an orthogonal projection on a hyperplane in a Poincaré disc (Left) and an Euclidean plane (Right).

$$f^c_{\boldsymbol{a},\boldsymbol{p}}(\boldsymbol{z}) = \mathrm{sign}(\langle\boldsymbol{a},\log^c_{\boldsymbol{p}}(\boldsymbol{z})\rangle_{\boldsymbol{p}})\,\|\boldsymbol{a}\|_{\boldsymbol{p}}\,d^c_p(\boldsymbol{z},H^c_{\boldsymbol{a},\boldsymbol{p}})$$

with $H^c_{\boldsymbol{a},\boldsymbol{p}} = \{\boldsymbol{z} \in \mathbb{B}^d_c \mid \langle \boldsymbol{a}, \log^c_{\boldsymbol{p}}(\boldsymbol{z}) \rangle = 0\} = \exp^c_{\boldsymbol{p}}(\{\boldsymbol{a}\}^{\perp})$. A closed-formed expression for the distance $d^c_p(\boldsymbol{z}, H^c_{\boldsymbol{a},\boldsymbol{p}})$ was also derived, $d^c_p(\boldsymbol{z}, H^c_{\boldsymbol{a},\boldsymbol{p}}) = \frac{1}{\sqrt{c}} \sinh^{-1}\left( \frac{2\sqrt{c}|\langle -\boldsymbol{p} \oplus_c \boldsymbol{z}, a \rangle|}{(1-c\|-\boldsymbol{p} \oplus_c \boldsymbol{z}\|^2)\|\boldsymbol{a}\|} \right)$. The hyperplane decision boundary $H^c_{\boldsymbol{a},\boldsymbol{p}}$ is called *gyroplane* and is a semi-hypersphere orthogonal to the Poincaré ball's boundary as illustrated on Figure 4. The decoder's first layer, called *gyroplane* layer, is chosen to be a concatenation of such operators, which are then composed with a standard feed-forward neural network.

**Encoder**   The encoder $g_\phi$ outputs a Fréchet mean $\boldsymbol{\mu} \in \mathbb{B}^d_c$ and a distortion $\sigma \in \mathbb{R}^+_*$ which parametrise the hyperbolic variational posterior. The Fréchet mean $\boldsymbol{\mu}$ is obtained as the image of the exponential map $\exp^c_{\boldsymbol{0}}$, and the distortion $\sigma$ through a *softplus* function.

## 3.3   Training

We follow a standard variational approach by deriving a lower bound on the marginal likelihood. The ELBO is optimised via an unbiased Monte Carlo (MC) estimator thanks to the reparametrisable sampling schemes that we introduce for both hyperbolic normal distributions.

**Objective**   The *evidence lower bound* (ELBO) can readily be extended to Riemannian latent spaces by applying Jensen's inequality w.r.t. $d\mathcal{M}$ (see Appendix A)

$$\log p(\boldsymbol{x}) \geq \mathcal{L}_\mathcal{M}(\boldsymbol{x}; \theta, \phi) \triangleq \int_\mathcal{M} \ln\left( \frac{p_\theta(\boldsymbol{x}|\boldsymbol{z})p(\boldsymbol{z})}{q_\phi(\boldsymbol{z}|\boldsymbol{x})} \right) q_\phi(\boldsymbol{z}|\boldsymbol{x}) \, d\mathcal{M}(\boldsymbol{z}).$$

Densities have been introduced earlier in Equations 1 and 2.

**Reparametrisation**   In the Euclidean setting, by working in polar coordinates, an isotropic normal distribution centred at $\boldsymbol{\mu}$ can be described by a directional vector $\boldsymbol{\alpha}$ uniformly distributed on the hypersphere and a univariate radius $r = d_E(\boldsymbol{\mu}, \boldsymbol{z})$ following a $\chi$-distribution. In the Poincaré ball we can rely on a similar representation, through a *hyperbolic polar* change of coordinates, given by

$$\boldsymbol{z} = \exp^c_{\boldsymbol{\mu}}\left( G(\boldsymbol{\mu})^{-\frac{1}{2}} \boldsymbol{v} \right) = \exp^c_{\boldsymbol{\mu}}\left( \frac{r}{\lambda^c_{\boldsymbol{\mu}}} \boldsymbol{\alpha} \right) \quad (3)$$

with $\boldsymbol{v} = r\boldsymbol{\alpha}$ and $r = d^c_p(\boldsymbol{\mu}, \boldsymbol{z})$. The direction $\boldsymbol{\alpha}$ is still uniformly distributed on the hypersphere and for the *wrapped normal*, the radius $r$ is still $\chi$-distributed, while for the *Riemannian normal* its density $\rho^\mathrm{R}(r)$ is given by (derived in Appendix B.4.1)

---

**Algorithm 1** Hyperbolic normal sampling scheme

---

**Require:** $\boldsymbol{\mu}, \sigma^2$, dimension $d$, curvature $c$
  **if** Wrapped normal **then** $\boldsymbol{v} \sim \mathcal{N}(\boldsymbol{0}_d, \sigma^2)$
  **else if** Riemannian normal **then**
      Let $g$ be a piecewise exponential proposal
      **while** sample $r$ not accepted **do**
          Propose $r \sim g(\cdot)$, $u \sim \mathcal{U}([0,1])$
          **if** $u < \frac{\rho^\mathrm{R}(r)}{g(r)}$ **then**  Accept sample $r$
      Sample direction $\boldsymbol{\alpha} \sim \mathcal{U}(\mathbb{S}^{d-1})$
      $\boldsymbol{v} \leftarrow r\boldsymbol{\alpha}$
  Return $\boldsymbol{z} = \exp^c_{\boldsymbol{\mu}}\left( \boldsymbol{v}/\lambda^c_{\boldsymbol{\mu}} \right)$

---

$$\rho^\mathrm{W}(r) \propto \mathbb{1}_{\mathbb{R}_+}(r) \, e^{-\frac{r^2}{2\sigma^2}} r^{d-1}, \quad \rho^\mathrm{R}(r) \propto \mathbb{1}_{\mathbb{R}_+}(r) e^{-\frac{r^2}{2\sigma^2}} \left( \frac{\sinh(\sqrt{c}r)}{\sqrt{c}} \right)^{d-1}.$$

The latter density $\rho^\mathrm{R}(r)$ can efficiently be sampled via rejection sampling with a piecewise exponential distribution proposal. This makes use of its log-concavity. The *Riemannian normal* sampling scheme is not directly affected by dimensionality since the radius is a *one-dimensional* random variable. Full sampling schemes are described in Algorithm 1, and in Appendices B.4.1 and B.4.2.

**Gradients**   Gradients $\nabla_{\boldsymbol{\mu}} \boldsymbol{z}$ can straightforwardly be computed thanks to the exponential map reparametrisation (Eq 3), and gradients w.r.t. the dispersion $\nabla_\sigma \boldsymbol{z}$ are readily available for the *wrapped normal*. For the *Riemannian normal*, we additionally rely on an implicit reparametrisation (Figurnov et al., 2018) of $\rho^\mathrm{R}$ via its cdf $F^\mathrm{R}(r; \sigma)$.

**Optimisation**   Parameters of the model living in the Poincaré ball are parametrised via the exponential mapping: $\phi_i = \exp^c_{\boldsymbol{0}}(\phi^0_i)$ with $\phi^0_i \in \mathbb{R}^m$, so we can make use of usual optimisation schemes. Alternatively, one could directly optimise such manifold parameters with manifold gradient descent schemes (Bonnabel, 2013).

# 4 Related work

**Hierarchical models** The Bayesian Nonparametric literature has a rich history of explicitly modelling the hierarchical structure of data (Teh et al., 2008; Heller and Ghahramani, 2005; Griffiths et al., 2004; Ghahramani et al., 2010; Larsen et al., 2001; Salakhutdinov et al., 2011). The discrete nature of trees used in such models makes learning difficult, whereas performing optimisation in a continuous hyperbolic space is an attractive alternative. Such an approach has been empirically and theoretically shown to be useful for graphs and word embeddings (Nickel and Kiela, 2017, 2018; Chamberlain et al., 2017; Sala et al., 2018; Tifrea et al., 2019).

**Distributions on manifold** Probability measures defined on manifolds are of interest to model uncertainty of data living (either intrinsically or assumed to) on such spaces, e.g. directional statistics (Ley and Verdebout, 2017; Mardia and Jupp, 2009). Pennec (2006) introduced a maximum entropy generalisation of the normal distribution, often referred to as *Riemannian normal*, which has been used for maximum likelihood estimation in the Poincaré half-plane (Said et al., 2014) and on the hypersphere (Hauberg, 2018). Another class of manifold probability measures are *wrapped* distributions, i.e. push-forward of distributions defined on a tangent space, often along the exponential map. They have recently been used in auto-encoder frameworks on the *hyperboloid* model (of hyperbolic geometry) (Grattarola et al., 2019; Nagano et al., 2019) and on Lie groups (Falorsi et al., 2018). Rey et al. (2019); Li et al. (2019) proposed to parametrise a variational family through a Brownian motion on manifolds such as spheres, tori, projective spaces and $SO(3)$.

**VAEs with Riemannian latent manifold** VAEs with non Euclidean latent space have been recently introduced, such as Davidson et al. (2018) making use of hyperspherical geometry and Falorsi et al. (2018) endowing the latent space with a SO(3) group structure. Concurrent work considers endowing auto-encoders (AEs) with a hyperbolic latent space. Grattarola et al. (2019) introduces a constant curvature manifold (CCM) (i.e. hyperspherical, Euclidean and hyperboloid) latent space within an adversarial auto-encoder framework. However, the encoder and decoder are not designed to explicitly take into account the latent space geometry. Ovinnikov (2018) recently proposed to endow a VAE latent space with a Poincaré ball model. They choose a Wasserstein Auto-Encoder framework (Tolstikhin et al., 2018) because they could not derive a closed-form solution of the ELBO's entropy term. We instead rely on a MC estimate of the ELBO by introducing a novel reparametrisation of the Riemannian normal. They discuss the *Riemannian* normal distribution, yet they make a number of heuristic approximations for sampling and reparametrisation. Also, Nagano et al. (2019) propose using a *wrapped* normal distribution to model uncertainty on the *hyperboloid* model of hyperbolic space. They derive its density and a reparametrisable sampling scheme, allowing such a distribution to be used in a variational learning framework. They apply this *wrapped* normal distribution to stochastically embed graphs and to parametrise the variational family in VAEs. Ovinnikov (2018) and Nagano et al. (2019) rely on a standard feed-forward decoder architecture, which does not take into account the hyperbolic geometry.

# 5 Experiments

We implemented our model and ran our experiments within the automatic differentiation framework PyTorch (Paszke et al., 2017). We open-source our code for reproducibility and to benefit the community [1]. Experimental details are fully described in Appendix C.

## 5.1 Branching diffusion process

We assess our modelling assumption on data generated from a branching diffusion process which explicitly incorporate hierarchical structure. Nodes $\boldsymbol{y}_i \in \mathbb{R}^n$ are normally distributed with mean given by their parent and with unit variance. Models are trained on a noisy vector representations $(\boldsymbol{x}_1, \dots, \boldsymbol{x}_N)$, hence do not have access to the true hierarchical representation. We train several $\mathcal{P}^c$-VAEs with increasing curvatures, along with a vanilla $\mathcal{N}$-VAE as a baseline. Table 1 shows that the $\mathcal{P}^c$-VAE outperforms its Euclidean counterpart in terms of test marginal likelihood. As expected, we observe that the performance of the $\mathcal{N}$-VAE is recovered as the curvature $c$ tends to zero. Also, we notice that increasing the prior distribution distortion $\sigma_0$ helps embeddings lie closer to the border,

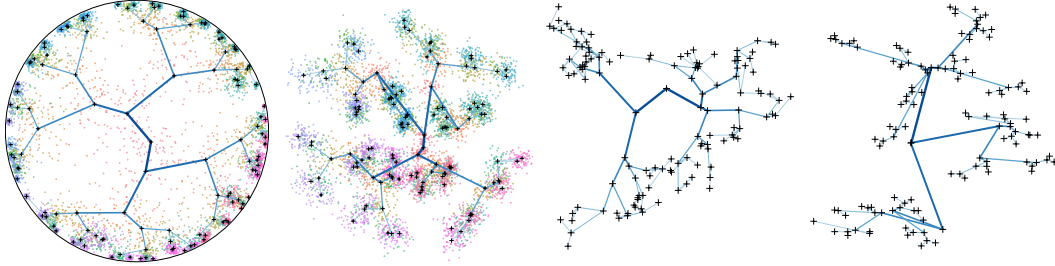

Figure 5: Latent representations learned by – $\mathcal{P}^1$-VAE (Leftmost), $\mathcal{N}$-VAE (Center-Left), PCA (Center-Right) and GPLVM (Rightmost) trained on synthetic dataset. Embeddings are represented by black crosses, and colour dots are posterior samples. Blue lines represent true hierarchy.

and as a consequence improved generalisation performance. Figure 5 represents latent embeddings for $\mathcal{P}^1$-VAE and $\mathcal{N}$-VAE, along with two embedding baselines: principal component analysis (PCA) and a Gaussian process latent variable model (GPLVM). A hierarchical structure is somewhat learned by all models, yet $\mathcal{P}^c$-VAE's latent representation is the least distorted.

Table 1: Negative test marginal likelihood estimates $\mathcal{L}_{\text{IWAE}}$ (Burda et al., 2015) (computed with 5000 samples) on the synthetic dataset. 95% confidence intervals are computed over 20 trainings.

| | $\sigma_0$ | Models | | | | | |
| | | $\mathcal{N}$-VAE | $\mathcal{P}^{0.1}$-VAE | $\mathcal{P}^{0.3}$-VAE | $\mathcal{P}^{0.8}$-VAE | $\mathcal{P}^{1.0}$-VAE | $\mathcal{P}^{1.2}$-VAE |
|---|---|---|---|---|---|---|---|
| $\mathcal{L}_{\text{IWAE}}$ | 1 | $57.1_{\pm 0.2}$ | $57.1_{\pm 0.2}$ | $57.2_{\pm 0.2}$ | $56.9_{\pm 0.2}$ | $56.7_{\pm 0.2}$ | $56.6_{\pm 0.2}$ |
| $\mathcal{L}_{\text{IWAE}}$ | 1.7 | $57.0_{\pm 0.2}$ | $56.8_{\pm 0.2}$ | $56.6_{\pm 0.2}$ | $55.9_{\pm 0.2}$ | $55.7_{\pm 0.2}$ | $\mathbf{55.6}_{\pm 0.2}$ |

## 5.2 Mnist digits

The MNIST (LeCun and Cortes, 2010) dataset has been used in the literature for hierarchical modelling (Salakhutdinov et al., 2011; Saha et al., 2018). One can view the natural clustering in MNIST images as a hierarchy with each of the 10 classes being internal nodes of the hierarchy. We empirically assess whether our model can take advantage of such simple underlying hierarchical structure, first by measuring its generalisation capacity via the test marginal log-likelihood. Table 2 shows that our model outperforms its Euclidean counterpart, especially for low latent dimension. This can be interpreted through an information bottleneck perspective; as the latent dimensionality increases, the pressure on the embeddings quality decreases, hence the gain from the hyperbolic geometry is reduced (as observed by Nickel and Kiela (2017)). Also, by using the *Riemannian normal* distribution, we achieve slightly better results than with the *wrapped normal*.

Table 2: Negative test marginal likelihood estimates computed with 5000 samples. 95% confidence intervals are computed over 10 runs. * indicates numerically unstable settings.

| | | Dimensionality | | | |
| | c | 2 | 5 | 10 | 20 |
|---|---|---|---|---|---|
| $\mathcal{N}$**-VAE** | (0) | $144.5_{\pm 0.4}$ | $114.7_{\pm 0.1}$ | $100.2_{\pm 0.1}$ | $97.6_{\pm 0.1}$ |
| | 0.1 | $143.9_{\pm 0.5}$ | $115.5_{\pm 0.3}$ | $100.2_{\pm 0.1}$ | $97.2_{\pm 0.1}$ |
| $\mathcal{P}$**-VAE (Wrapped)** | 0.2 | $144.2_{\pm 0.5}$ | $115.3_{\pm 0.3}$ | $100.0_{\pm 0.1}$ | $97.1_{\pm 0.1}$ |
| | 0.7 | $143.8_{\pm 0.6}$ | $115.1_{\pm 0.3}$ | $100.2_{\pm 0.1}$ | $97.5_{\pm 0.1}$ |
| | 1.4 | $144.0_{\pm 0.6}$ | $114.7_{\pm 0.1}$ | $100.7_{\pm 0.1}$ | $98.0_{\pm 0.1}$ |
| | 0.1 | $143.7_{\pm 0.6}$ | $115.2_{\pm 0.2}$ | $99.9_{\pm 0.1}$ | $\mathbf{97.0}_{\pm 0.1}$ |
| $\mathcal{P}$**-VAE (Riemannian)** | 0.2 | $143.8_{\pm 0.4}$ | $114.7_{\pm 0.3}$ | $\mathbf{99.7}_{\pm 0.1}$ | $97.4_{\pm 0.1}$ |
| | 0.7 | $143.1_{\pm 0.4}$ | $\mathbf{114.1}_{\pm 0.2}$ | $101.2_{\pm 0.2}$ | * |
| | 1.4 | $\mathbf{142.5}_{\pm 0.4}$ | $115.5_{\pm 0.3}$ | * | * |

We conduct an ablation study to assess the usefulness of the *gyroplane* layer introduced in Section 3.2. To do so we estimate the test marginal log-likelihood for different choices of decoder. We select a multi-layer perceptron (MLP) to be the baseline decoder. We additionally compare to a MLP pre-composed by $\log_{\mathbf{0}}$, which can be seen as a linearisation of the space around the centre of the ball. Figure 6 shows the relative performance improvement of decoders over the MLP baseline w.r.t. the latent space dimension. We observe that *linearising* the input of a MLP through the logarithm map slightly improves generalisation, and that using a *gyroplane* layer as the first layer of the decoder additionally improves generalisation. Yet, these performance gains appear to decrease as the latent dimensionality increases.

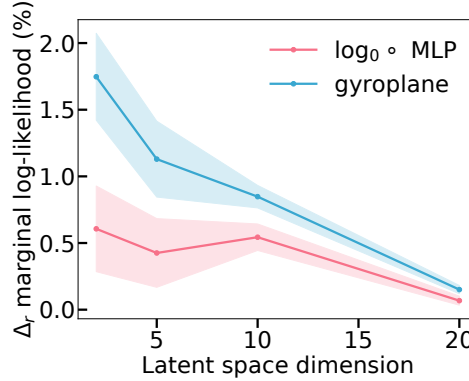

Figure 6: Decoder ablation study on MNIST with *wrapped* normal $\mathcal{P}^1$-VAE. Baseline decoder is a MLP.

Second, we explore the learned latent representations of the trained $\mathcal{P}$-VAE and $\mathcal{N}$-VAE models shown in Figure 7. Qualitatively our $\mathcal{P}$-VAE produces a clearer partitioning of the digits, in groupings of $\{4, 7, 9\}$, $\{0, 6\}$, $\{2, 3, 5, 8\}$ and $\{1\}$, with right-slanting $\{5, 8\}$ being placed separately from the non-slanting ones. Recall that distances increase towards the edge of the Poincaré ball. We quantitatively assess the quality of the embeddings by training a classifier predicting labels. Table 3 shows that the embeddings learned by our $\mathcal{P}$-VAE model yield on average an $2\%$ increase in accuracy over the digits. The full confusion matrices are shown in Figure 12 in Appendix.

Table 3: Per digit accuracy of a classifier trained on the learned latent 2-d embeddings. Results are averaged over 10 sets of embeddings and 5 classifier trainings.

| Digits | 0 | 1 | 2 | 3 | 4 | 5 | 6 | 7 | 8 | 9 | Avg |
|---|---|---|---|---|---|---|---|---|---|---|---|
| $\mathcal{N}$-VAE | 89 | 97 | 81 | 75 | 59 | 43 | 89 | **78** | 68 | **57** | 73.6 |
| $\mathcal{P}^{1.4}$-VAE | **94** | 97 | **82** | **79** | **69** | **47** | **90** | 77 | 68 | 53 | **75.6** |

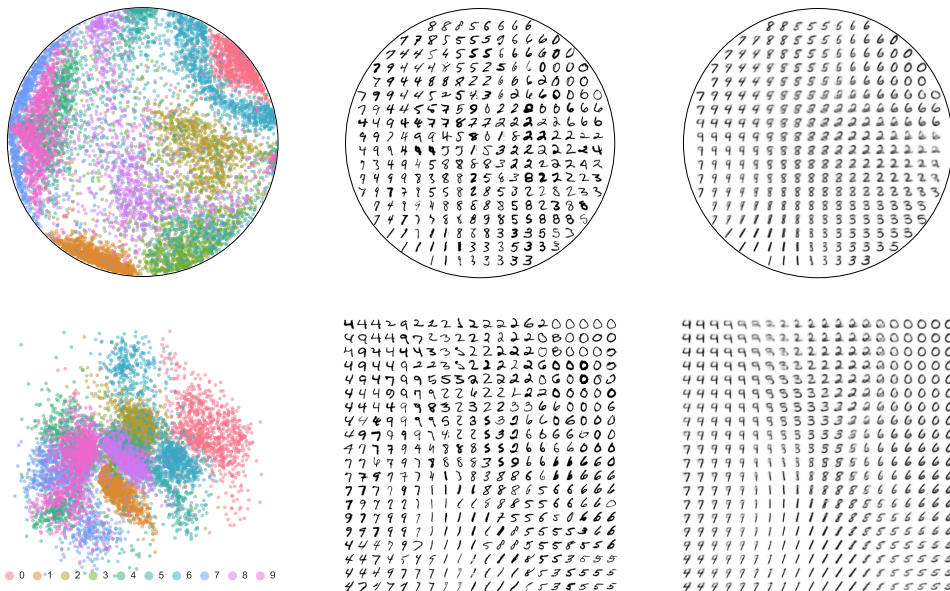

Figure 7: MNIST Posteriors mean (Left) sub-sample of digit images associated with posteriors mean (Middle) Model samples (Right) – for $\mathcal{P}^{1.4}$-VAE (Top) and $\mathcal{N}$-VAE (Bottom).

## 5.3  Graph embeddings

We evaluate the performance of a variational graph auto-encoder (VGAE) (Kipf and Welling, 2016) with Poincaré ball latent space for link prediction in networks. Edges in complex networks can typically be explained by a latent hierarchy over the nodes (Clauset et al., 2008). We believe the Poincaré ball latent space should help in terms of generalisation. We demonstrate these capabilities on three network datasets: a graph of Ph.D. advisor-advisee relationships (Nooy et al., 2011), a phylogenetic tree expressing genetic heritage (Hofbauer et al., 2016; Sanderson and Eriksson, 1994) and a biological set representing disease relationships (Goh et al., 2007; Rossi and Ahmed, 2015).

We follow the VGAE model, which maps the adjacency matrix $\boldsymbol{A}$ to node embeddings $\boldsymbol{Z}$ through a graph convolutional network (GCN), and reconstructs $\boldsymbol{A}$ by predicting edge probabilities from the node embeddings. In order to take into account the latent space geometry, we parametrise the probability of an edge by $p(\boldsymbol{A}_{ij} = 1 | \boldsymbol{z}_i, \boldsymbol{z}_j) = 1 - \tanh(d_{\mathcal{M}}(\boldsymbol{z}_i, \boldsymbol{z}_j)) \in (0, 1]$ with $d_{\mathcal{M}}$ the latent geodsic metric. We use a *Wrapped* Gaussian prior and variational posterior for the $\mathcal{P}^1$-VAE.

We set the latent dimension to $5$. We follow the training and evaluation procedures introduced in Kipf and Welling (2016). Models are trained on an incomplete adjacency matrix where some of the edges have randomly been removed. A test set is formed from previously removed edges and an equal number of randomly sampled pairs of unconnected nodes. We report in Table 4 the *area under the ROC curve* (AUC) and *average precision* (AP) evaluated on the test set. It can be observed that the $\mathcal{P}$-VAE performs better than its Euclidean counterpart in terms of generalisation to unseen edges.

Table 4: Results on network link prediction. 95% confidence intervals are computed over 40 runs.

|  | Phylogenetic | | CS PhDs | | Diseases | |
|---|---|---|---|---|---|---|
|  | **AUC** | **AP** | **AUC** | **AP** | **AUC** | **AP** |
| $\mathcal{N}$-**VAE** | $54.2_{\pm 2.2}$ | $54.0_{\pm 2.1}$ | $56.5_{\pm 1.1}$ | $56.4_{\pm 1.1}$ | $89.8_{\pm 0.7}$ | $91.8_{\pm 0.7}$ |
| $\mathcal{P}$-**VAE** | $\mathbf{59.0}_{\pm \mathbf{1.9}}$ | $55.5_{\pm 1.6}$ | $\mathbf{59.8}_{\pm 1.2}$ | $56.7_{\pm 1.2}$ | $\mathbf{92.3}_{\pm 0.7}$ | $\mathbf{93.6}_{\pm 0.5}$ |

# 6  Conclusion

In this paper we have explored VAEs with a Poincaré ball latent space. We gave a thorough treatment of two canonical – *wrapped* and *maximum entropy* – normal generalisations on that space, and a rigorous analysis of the difference between the two. We derived the necessary ingredients for training such VAEs, namely efficient and reparametrisable sampling schemes, along with probability density functions for these two distributions. We introduced a decoder architecture explicitly taking into account the hyperbolic geometry, and empirically showed that it is crucial for the hyperbolic latent space to be useful. We empirically demonstrated that endowing a VAE with a Poincaré ball latent space can be beneficial in terms of model generalisation and can yield more interpretable representations if the data has hierarchical structure.

There are a number of interesting future directions. There are many models of hyperbolic geometry, and several have been considered in a gradient-based setting. Yet, it is still unclear which models should be preferred and which of their properties matter. Also, it would be useful to consider principled ways of assessing whether a given dataset has an underlying hierarchical structure, in the same way that topological data analysis (Pascucci et al., 2011) attempts to discover the topologies that underlie datasets.

**Acknowledgments** We are extremely grateful to Adam Foster, Phillipe Gagnon and Emmanuel Chevallier for their help. EM, YWT's research leading to these results received funding from the European Research Council under the European Union's Seventh Framework Programme (FP7/2007-2013) ERC grant agreement no. 617071 and they acknowledge Microsoft Research and EPSRC for funding EM's studentship, and EPSRC grant agreement no. EP/N509711/1 for funding CL's studentship.

## Footnotes

[1] https://github.com/emilemathieu/pvae

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
