[Supplementary Material]

## A  Evidence Lower Bound

The ELBO can readily be extended for Riemannian latent spaces by applying Jensen's inequality w.r.t. the metric induced measure $d\mathcal{M}$ which yield

$$
\begin{aligned}
\ln p(\boldsymbol{x}) = \ln \int_{\mathcal{Z}=\mathcal{M}} p_\theta(\boldsymbol{x}, \boldsymbol{z}) d\mathcal{M}(\boldsymbol{z}) &= \ln \int_{\mathcal{M}} p_\theta(\boldsymbol{x}|\boldsymbol{z}) p(\boldsymbol{z}) d\mathcal{M}(\boldsymbol{z}) \\
&= \ln \int_{\mathcal{M}} \frac{p_\theta(\boldsymbol{x}|\boldsymbol{z}) p(\boldsymbol{z})}{q_\phi(\boldsymbol{z}|\boldsymbol{x})} q_\phi(\boldsymbol{z}|\boldsymbol{x}) d\mathcal{M}(\boldsymbol{z}) \\
&\geq \int_{\mathcal{M}} \ln \frac{p_\theta(\boldsymbol{x}|\boldsymbol{z}) p(\boldsymbol{z})}{q_\phi(\boldsymbol{z}|\boldsymbol{x})} q_\phi(\boldsymbol{z}|\boldsymbol{x}) d\mathcal{M}(\boldsymbol{z}) \\
&= \int_{\mathcal{M}} \left[ \ln p_\theta(\boldsymbol{x}|\boldsymbol{z}) - \ln p(\boldsymbol{z}) - \ln q_\phi(\boldsymbol{z}|\boldsymbol{x}) \right] q_\phi(\boldsymbol{z}|\boldsymbol{x}) \, d\mathcal{M}(\boldsymbol{z}) \\
&= \mathbb{E}_{\boldsymbol{z} \sim q_\phi(\cdot|\boldsymbol{x})\mathcal{M}(\cdot)} \left[ \ln p_\theta(\boldsymbol{x}|\boldsymbol{z}) + \ln p(\boldsymbol{z}) - \ln q_\phi(\boldsymbol{z}|\boldsymbol{x}) \right] \\
&\triangleq \mathcal{L}_{\mathcal{M}}(\boldsymbol{x}; \theta, \phi) \\
&\approx \sum_k \ln p_\theta(\boldsymbol{x}|\boldsymbol{z}^k) + \ln p(\boldsymbol{z}^k) - \ln q_\phi(\boldsymbol{z}^k|\boldsymbol{x}), \quad \boldsymbol{z}^k \sim q_\phi(\cdot|\boldsymbol{x})\sqrt{|G(\cdot)|}
\end{aligned}
$$

## B  Hyperbolic normal distributions

In this section, we first review some canonical generalisation of the normal distributions to Riemannian manifolds, and then introduce in more details the *Riemannian* and *wrapped* normal distributions on the Poincaré ball. Finally, we give architecture and training details about the conducted experiments.

### B.1  Probability measures on Riemannian manifolds

Probability measures and random vectors can intrinsically be defined on Riemannian manifolds so as to model uncertainty on non-flat spaces (Pennec, 2006). The Riemannian metric $G(\boldsymbol{z})$ induces an infinitesimal volume element on each tangent space $\mathcal{T}_{\boldsymbol{z}}\mathcal{M}$, and thus a measure on the manifold,

$$
d\mathcal{M}(\boldsymbol{z}) = \sqrt{|G(\boldsymbol{z})|} d\boldsymbol{z}, \tag{4}
$$

with $d\boldsymbol{z}$ being the Lebesgue measure. Random variables $\boldsymbol{z} \in \mathcal{M}$ would naturally be characterised by the Radon-Nikodym derivative of a measure $\nu$ w.r.t. the Riemannian measure $d\mathcal{M}(\cdot)$ (assuming absolute continuity)

$$
f(\boldsymbol{z}) = \frac{d\nu(\boldsymbol{z})}{d\mathcal{M}(\boldsymbol{z})}.
$$

Since the normal distribution plays such a canonical role in statistics, generalising it to manifold is of interest. Given a Fréchet expectation $\boldsymbol{\mu} \in \mathcal{M}$ – defined as minimisers of $\int_{\mathcal{M}} d_{\mathcal{M}}(\boldsymbol{\mu}, \boldsymbol{z})^2 p(\boldsymbol{z}) d\mathcal{M}(\boldsymbol{z})$ – and a dispersion parameter $\sigma > 0$ (generally not equal to the standard deviation), several properties ought to be verified by such generalised normal distributions. Such a distribution should tend towards a delta function at $\boldsymbol{\mu}$ when $\sigma \to 0$ and to an (improper for non-compact) uniform distribution when $\sigma \to \infty$. Also, as the curvature tends to 0, one should recover the vanilla normal distribution. Hereby, we review canonical generalisations of the normal distribution, which have different theoretical and computational advantages.

**Maximum entropy normal**  The property that Pennec (2006) takes for granted is the maximization of the entropy given a mean and a covariance matrix, yielding in the isotropic setting

$$
\frac{d\nu^{\mathrm{R}}(\boldsymbol{z}|\boldsymbol{\mu}, \sigma^2)}{d\mathcal{M}(\boldsymbol{z})} = \mathcal{N}_{\mathcal{M}}^{\mathrm{R}}(\boldsymbol{z}|\boldsymbol{\mu}, \sigma^2) = \frac{1}{Z^{\mathrm{R}}} \exp\left( -\frac{d_{\mathcal{M}}(\boldsymbol{\mu}, \boldsymbol{z})^2}{2\sigma^2} \right), \tag{5}
$$

with $d_{\mathcal{M}}$ being the Riemannian distance on the manifold induced by the tensor metric. Such a formulation – sometimes referred as *Riemannian Normal* distribution – is used by Said et al. (2014) in the Poincaré half-plane, or by Hauberg (2018) in the hypersphere $\mathbb{S}^d$. Sampling from such distributions and computing the normalising constant – especially in the anisotropic setting – is usually challenging.

**Wrapped normal** Another generalisation is defined by taking the image by the exponential map of a Gaussian distribution on the tangent space centered at the mean value. Such a distribution has been referred in literature as *wrapped*, *push-forward*, *exp-map* or *tangential* normal distribution. Sampling is therefore straightforward. The pdf is then readily available through the change of variable formula if one can compute the Jacobian of the exponential map (or its inverse). Hence such a distribution is attractive from a computational perspective. Grattarola et al. (2019) and Nagano et al. (2019) rely on such a distribution defined on the hyperboloid model. Wrapped distributions are often encountered in the *directional* statistics (Ley and Verdebout, 2017; Hauberg, 2018).

**Restricted normal** What is more, for sub-manifolds of $\mathbb{R}^n$, one can consider the restriction of a normal distribution pdf to the manifold. This yields the Von Mises distribution on $\mathbb{S}^1$ and the Von Mises-Fisher distribution on $\mathbb{S}^d$ (Hauberg, 2018) and the Stiefel manifold.

**Diffusion normal** Yet another generalisation arises by defining the normal pdf through the *heat kernel*, or fundamental solution of the heat equation, $K : \mathbb{R}^+ \times \mathcal{M} \times \mathcal{M} \to \mathcal{M}$,

$$\mathcal{N}_{\mathcal{M}}^{\Delta}(\boldsymbol{z}|\boldsymbol{\mu}, \sigma^2) = K(\sigma^2/2, \boldsymbol{\mu}, \boldsymbol{z}). \tag{6}$$

See for instance Hsu (2008) for an introduction of Brownian motion on Riemannian manifolds and Paeng (2011) for conditions on existence and uniqueness of the kernel. Sampling amounts to simulating a Brownian motion, which may be challenging for non sub-manifolds of $\mathbb{R}^n$. Closed form solutions of the heat kernel is available for some manifolds such as spheres or flat tori, otherwise numerical approximations can be used. Such a distribution has been used in a VAE setting (Rey et al., 2019; Li et al., 2019).

**Other than normal distributions** Of course one needs not to restrict itself to generalisations of the normal distribution. For instance, one could consider a wrapped spherical Student-t as $\boldsymbol{z} \sim \exp_{\boldsymbol{\mu}\#} S_t(0, \nu)$ or a Riemannian Student-t with density proportional to $\left(1 + d_{\mathcal{M}}(\boldsymbol{z}, \boldsymbol{\mu})^2/\nu\right)^{(-\nu+1)/2}$ (by making sure that this density is $d\mathcal{M}$-integrable).

## B.2 Hyperbolic polar coordinates

In this subsection, we review the hyperbolic polar change of coordinates allowing us to reparametrise hyperbolic normal distributions in a similar fashion than the Box–Muller transform (Box and Muller, 1958).

**Polar coordinates** Euclidean polar coordinates, express points $\boldsymbol{z} \in \mathbb{R}^d$ through a radius $r \geq 0$ and a direction $\boldsymbol{\alpha} \in \mathbb{S}^{d-1}$ such that $\boldsymbol{z} = r\boldsymbol{\alpha}$. Yet, one could choose another *pole* (or *reference point*) $\boldsymbol{\mu} \neq \boldsymbol{0}$ such that $\boldsymbol{z} = \boldsymbol{\mu} + r\boldsymbol{\alpha}$. Consequently, $r = d_E(\boldsymbol{\mu}, \boldsymbol{z})$. An analogous change of variables can also be constructed in Riemannian manifolds relying on the exponential map instead of the addition operator. Given a *pole* $\boldsymbol{\mu} \in \mathbb{B}_c^d$, the point of hyperbolic polar coordinates $\boldsymbol{z} = (r, \boldsymbol{\alpha})$ is defined as $\boldsymbol{z} = \gamma(r)$, with $r = d_p^c(\boldsymbol{\mu}, \boldsymbol{z})$ and $\gamma : \mathbb{R}^+ \to \mathbb{B}_c^d$ a curve such that $\gamma'(0) = \boldsymbol{\alpha} \in \mathbb{S}^{d-1}$. Hence $\boldsymbol{z} = \exp_{\boldsymbol{\mu}}^c\left(\frac{r}{\lambda_{\boldsymbol{\mu}}^c}\boldsymbol{\alpha}\right)$ since $d_p^c(\boldsymbol{\mu}, \boldsymbol{z}) = \|\ln_{\boldsymbol{\mu}}^c(x)\|_{\boldsymbol{\mu}} = \|\frac{r}{\lambda_{\boldsymbol{\mu}}^c}\boldsymbol{\alpha}\|_{\boldsymbol{\mu}} = r$.

**Tensor metric** We derive below the expression of the Poincaré ball metric in such hyperbolic polar coordinate, for the specific setting where $\boldsymbol{\mu} = \boldsymbol{0}$: $\boldsymbol{z} = \exp_{\boldsymbol{0}}^c(\frac{r}{2}\boldsymbol{\alpha})$. Switching to Euclidean polar coordinate we get

$$ds_{\mathbb{B}_c^d}^2 = (\lambda_{\boldsymbol{z}}^c)^2(dz_1^2 + \cdots + dz_d^2) = \frac{4}{(1 - c\|x\|^2)^2}d\boldsymbol{z}^2$$

$$= \frac{4}{(1 - c\rho^2)^2}(d\rho^2 + \rho^2 ds_{\mathbb{S}^{d-1}}^2). \tag{7}$$

Let's define $r = d_p^c(\boldsymbol{0}, \boldsymbol{z}) = L(\gamma)$, with $\gamma$ being the geodesic joining $\boldsymbol{0}$ and $\boldsymbol{z}$. Since such a geodesic is the segment $[\boldsymbol{0}, \boldsymbol{z}]$, we have

$$r = \int_0^\rho \lambda_t^c dt = \int_0^\rho \frac{2}{1 - ct^2} dt = \int_0^{\sqrt{c}\rho} \frac{2}{1 - t^2} \frac{dt}{\sqrt{c}} = \frac{2}{\sqrt{c}} \tanh^{-1}(\sqrt{c}\rho).$$

Plugging $\rho = \frac{1}{\sqrt{c}}\tanh(\sqrt{c}\frac{r}{2})$ (and $d\rho = (1 - c\rho^2)/2dr$) into Eq 7 yields

$$ds^2_{\mathbb{B}^d_c} = \frac{4}{(1-c\rho^2)^2}\frac{1}{4}(1-c\rho^2)^2 dr^2 + \left(2\frac{\rho}{1-c\rho^2}\right)^2 ds^2_{\mathbb{S}^{d-1}}$$

$$= dr^2 + \left(2\frac{\frac{1}{\sqrt{c}}\tanh(\sqrt{c}\frac{r}{2})}{1 - c\left(\frac{1}{\sqrt{c}}\tanh(\sqrt{c}\frac{r}{2})\right)^2}\right)^2 ds^2_{\mathbb{S}^{d-1}}$$

$$= dr^2 + \left(\frac{1}{\sqrt{c}}\sinh(\sqrt{c}r)\right)^2 ds^2_{\mathbb{S}^{d-1}}. \tag{8}$$

The Euclidean line element is recovered when $c \to 0$

$$ds^2_{\mathbb{R}^d} = dr^2 + r^2 ds^2_{\mathbb{S}^{d-1}}. \tag{9}$$

In an appropriate orthonormal basis of $\mathcal{T}_{\boldsymbol{\mu}}\mathbb{B}^d_c$ the hyperbolic polar coordinate leads to the following expression of the matrix of the metric

$$G(\boldsymbol{z}) = \begin{pmatrix} 1 & 0 \\ 0 & \left(\frac{\sinh(\sqrt{c}r)}{\sqrt{c}r}\right)^2 \boldsymbol{I}_{d-1} \end{pmatrix}. \tag{10}$$

Hence, the density of the Riemannian measure with respect to the image of the Lebesgue measure of $\mathcal{T}_{\boldsymbol{\mu}}\mathbb{B}^d_c$ by $\exp^c_{\boldsymbol{\mu}}$ is given by

$$\sqrt{|G(\boldsymbol{z})|} = \left(\frac{\sinh(\sqrt{c}r)}{\sqrt{c}r}\right)^{d-1}. \tag{11}$$

This result holds for any *reference point* $\boldsymbol{\mu} \in \mathbb{B}^d_c$, with $r = d^c_p(\boldsymbol{\mu}, \boldsymbol{z})$, since the metric induced measure is invariant under the isometries of the manifold (i.e. Möbius transformations). This result can also be found in Chevallier et al. (2015); Said et al. (2014). Also, the fact that the line element $ds^2_{\mathbb{B}^d_c}$ and equivalently the metric $G$ only depends on the radius in hyperbolic polar coordinate, is a consequence of the hyperbolic space's isotropy.

**Integration** We now make use of the aforementioned hyperbolic polar coordinates to integrate functions following Said et al. (2014). The integral of a function $f : \mathbb{B}^d_c \to \mathbb{R}$ can be computed by using polar coordinates,

$$\int_{\mathbb{B}^d_c} f(\boldsymbol{z})d\mathcal{M}(\boldsymbol{z}) = \int_{\mathbb{B}^d_c} f(\boldsymbol{z})\sqrt{|G(\boldsymbol{z})|}\, d\boldsymbol{z}$$

$$= \int_{\mathcal{T}_{\boldsymbol{\mu}}\mathbb{B}^d_c \cong \mathbb{R}^d} f(\boldsymbol{v})\sqrt{|G(\boldsymbol{v})|}\, d\boldsymbol{v} \tag{12}$$

$$= \int_{\mathbb{R}_+}\int_{\mathbb{S}^{d-1}} f(r)\sqrt{|G(r)|}dr\, r^{d-1}\, ds_{\mathbb{S}^{d-1}}$$

$$= \int_{\mathbb{R}_+}\int_{\mathbb{S}^{d-1}} f(r)\left(\frac{\sinh(\sqrt{c}r)}{\sqrt{c}r}\right)^{d-1} dr\, r^{d-1}\, ds_{\mathbb{S}^{d-1}}$$

$$= \int_{\mathbb{R}_+}\int_{\mathbb{S}^{d-1}} f(r)\left(\frac{\sinh(\sqrt{c}r)}{\sqrt{c}}\right)^{d-1} dr\, ds_{\mathbb{S}^{d-1}}. \tag{13}$$

## B.3 Wrapped hyperbolic normal distribution on $\mathbb{B}^d_c$

**Anisotropic** The *wrapped normal* distribution considers a normal distribution in the tangent space $\mathcal{T}_{\boldsymbol{\mu}}\mathbb{B}^d_c$ being pushed-forward along the exponential map. One can obtain sampled as follow

$$\boldsymbol{z} = \exp^c_{\boldsymbol{\mu}}\left(G(\boldsymbol{\mu})^{-\frac{1}{2}}\boldsymbol{v}\right) = \exp^c_{\boldsymbol{\mu}}\left(\frac{\boldsymbol{v}}{\lambda^c_{\boldsymbol{\mu}}}\right), \text{ with } \boldsymbol{v} \sim \mathcal{N}(\cdot|\boldsymbol{0}, \Sigma). \tag{14}$$

Then, its density is given by

$$\mathcal{N}^{\mathrm{W}}_{\mathbb{B}^d_c}(\boldsymbol{z}|\boldsymbol{\mu},\Sigma) = \mathcal{N}\left(G(\boldsymbol{\mu})^{1/2}\log_{\boldsymbol{\mu}}(\boldsymbol{z})\;\middle|\;\boldsymbol{0},\Sigma\right)\left(\frac{\sqrt{c}\,d_p^c(\boldsymbol{\mu},\boldsymbol{z})}{\sinh(\sqrt{c}\,d_p^c(\boldsymbol{\mu},\boldsymbol{z}))}\right)^{d-1}$$

$$= \mathcal{N}\left(\lambda^c_{\boldsymbol{\mu}}\,\log_{\boldsymbol{\mu}}(\boldsymbol{z})\;\middle|\;\boldsymbol{0},\Sigma\right)\left(\frac{\sqrt{c}\,d_p^c(\boldsymbol{\mu},\boldsymbol{z})}{\sinh(\sqrt{c}\,d_p^c(\boldsymbol{\mu},\boldsymbol{z}))}\right)^{d-1} \tag{15}$$

with $G(\boldsymbol{\mu})^{1/2}$ the unique square-root matrix of $G(\boldsymbol{\mu})$ (thanks to the positive definiteness of the metric tensor). This can be shown by plugging this density as $f$ in Equation (12) with $\boldsymbol{v} = r\boldsymbol{\alpha} = \lambda^c_{\boldsymbol{\mu}}\log_{\boldsymbol{\mu}}(\boldsymbol{z})$, we get

$$\int_{\mathbb{B}^d_c}\mathcal{N}^{\mathrm{W}}_{\mathbb{B}^d_c}(\boldsymbol{z}|\boldsymbol{\mu},\Sigma)\,d\mathcal{M}(\boldsymbol{z}) = \int_{\mathcal{T}_{\boldsymbol{\mu}}\mathbb{B}^d_c\cong\mathbb{R}^d}\mathcal{N}\left(\boldsymbol{v}\mid\boldsymbol{0},\Sigma\right)\left(\frac{\sqrt{c}\,\|\boldsymbol{v}\|_2}{\sinh(\sqrt{c}\,\|\boldsymbol{v}\|_2)}\right)^{d-1}\sqrt{|G(\boldsymbol{v})|}\,d\boldsymbol{v}$$

$$= \int_{\mathbb{R}^d}\mathcal{N}\left(\boldsymbol{v}\mid\boldsymbol{0},\Sigma\right)\left(\frac{\sqrt{c}\,\|\boldsymbol{v}\|_2}{\sinh(\sqrt{c}\,\|\boldsymbol{v}\|_2)}\right)^{d-1}\left(\frac{\sinh(\sqrt{c}\,\|\boldsymbol{v}\|_2)}{\sqrt{c}\,\|\boldsymbol{v}\|_2}\right)^{d-1}\,d\boldsymbol{v}$$

$$= \int_{\mathbb{R}^d}\mathcal{N}\left(\boldsymbol{v}\mid\boldsymbol{0},\Sigma\right)\,d\boldsymbol{v}.$$

Figure 8: Anisotropic wrapped normal probability measures for Fréchet means $\boldsymbol{\mu}$ (red +), concentrations $\Sigma = \mathrm{diag}(\boldsymbol{\sigma})$ and $c = 1$.

**Isotropic**   In the isotropic setting, we therefore get

$$\int_{\mathbb{B}^d_c}\mathcal{N}^{\mathrm{W}}_{\mathbb{B}^d_c}(\boldsymbol{z}|\boldsymbol{\mu},\sigma^2)\,d\mathcal{M}(\boldsymbol{z}) = \int_{\mathbb{R}_+}\int_{\mathbb{S}^{d-1}}\frac{1}{Z^{\mathrm{R}}}e^{-\frac{r^2}{2\sigma^2}}r^{d-1}dr\,ds_{\mathbb{S}^{d-1}}. \tag{16}$$

The hyperbolic radius $r = d_p^c(\boldsymbol{\mu},\boldsymbol{z})$ consequently follows the usual $\chi$ distribution with density

$$\rho^{\mathrm{W}}(r) \propto \mathbb{1}_{\mathbb{R}_+}(r)\,e^{-\frac{r^2}{2\sigma^2}}r^{d-1}, \tag{17}$$

and the density of the *wrapped* normal given by

$$\mathcal{N}^{\mathrm{W}}_{\mathbb{B}^d_c}(\boldsymbol{z}|\boldsymbol{\mu},\sigma^2) = \frac{d\nu^{\mathrm{W}}(\boldsymbol{z}|\boldsymbol{\mu},\sigma^2)}{d\mathcal{M}(\boldsymbol{z})} = (2\pi\sigma^2)^{-d/2}\exp\left(-\frac{d_p^c(\boldsymbol{\mu},\boldsymbol{z})^2}{2\sigma^2}\right)\left(\frac{\sqrt{c}\,d_p^c(\boldsymbol{\mu},\boldsymbol{z})}{\sinh(\sqrt{c}\,d_p^c(\boldsymbol{\mu},\boldsymbol{z}))}\right)^{d-1}.$$

## B.4 Maximum entropy hyperbolic normal distribution on $\mathbb{B}_c^d$

Alternatively, by considering the maximum entropy generalisation of the normal distribution one gets (Pennec, 2006)

$$\mathcal{N}_{\mathbb{B}_c^d}^{\mathrm{R}}(\boldsymbol{z}|\boldsymbol{\mu},\sigma^2) = \frac{d\nu^{\mathrm{R}}(\boldsymbol{z}|\boldsymbol{\mu},\sigma^2)}{d\mathcal{M}(\boldsymbol{z})} = \frac{1}{Z^{\mathrm{R}}}\exp\left(-\frac{d_p^c(\boldsymbol{\mu},\boldsymbol{z})^2}{2\sigma^2}\right). \tag{18}$$

Such a pdf can be computed pointwise once $Z^{\mathrm{R}}$ is known, which we derive in Appendix B.4.3. Also, we observe that as $c$ and $\sigma$ get smaller (resp. bigger), the *Riemannian* normal pdf gets closer (resp. further) to the *wrapped* normal pdf.

### B.4.1 Reparametrisation

Plugging the *Riemannian* normal density as $f$ in Equation (13), with $r = d_p^c(\boldsymbol{\mu},\boldsymbol{z})$, we have

$$\int_{\mathbb{B}_c^d} \mathcal{N}_{\mathbb{B}_c^d}^{\mathrm{R}}(\boldsymbol{z}|\boldsymbol{\mu},\sigma^2)\,d\mathcal{M}(\boldsymbol{z}) = \int_{\mathbb{R}_+}\int_{\mathbb{S}^{d-1}} \frac{1}{Z^{\mathrm{R}}} e^{-\frac{r^2}{2\sigma^2}}\left(\frac{\sinh(\sqrt{c}r)}{\sqrt{c}}\right)^{d-1} dr\,ds_{\mathbb{S}^{d-1}}$$

$$= \frac{1}{Z^{\mathrm{R}}}\left(\int_{\mathbb{R}_+} e^{-\frac{r^2}{2\sigma^2}}\left(\frac{\sinh(\sqrt{c}r)}{\sqrt{c}}\right)^{d-1} dr\right)\left(\int_{\mathbb{S}^{d-1}} ds_{\mathbb{S}^{d-1}}\right) \tag{19}$$

Hence, samples $\boldsymbol{z} \sim \mathcal{N}_{\mathcal{M}}^{\mathrm{R}}(\boldsymbol{z}|\boldsymbol{\mu},\sigma^2)d\mathcal{M}(\boldsymbol{z})$ can be reparametrised as

$$\boldsymbol{z} = \exp_{\boldsymbol{\mu}}^c\left(\frac{r}{\lambda_{\boldsymbol{\mu}}^c}\boldsymbol{\alpha}\right) \tag{20}$$

with the direction $\boldsymbol{\alpha}$ being uniformly distributed on the hypersphere $\mathbb{S}^{d-1}$, i.e.

$$\boldsymbol{\alpha} \sim \mathcal{U}(\mathbb{S}^{d-1})$$

and the hyperbolic radius $r = d_p^c(\boldsymbol{\mu},\boldsymbol{z})$ distributed according to the following density (w.r.t the Lebesgue measure)

$$\rho^{\mathrm{R}}(r) = \frac{\mathbb{1}_{\mathbb{R}_+}(r)}{Z_r^{\mathrm{R}}} e^{-\frac{r^2}{2\sigma^2}}\left(\frac{\sinh(\sqrt{c}r)}{\sqrt{c}}\right)^{d-1}. \tag{21}$$

**Developed expression** By expanding the $\sinh$ term using the binomial formula, we get

$$\rho^{\mathrm{R}}(r) = \frac{\mathbb{1}_{\mathbb{R}_+}(r)}{Z_r^{\mathrm{R}}} e^{-\frac{r^2}{2\sigma^2}}\left(\frac{\sinh(\sqrt{c}r)}{\sqrt{c}}\right)^{d-1}$$

$$= \frac{\mathbb{1}_{\mathbb{R}_+}(r)}{Z_r^{\mathrm{R}}} e^{-\frac{r^2}{2\sigma^2}}\left(\frac{e^{\sqrt{c}r} - e^{-\sqrt{c}r}}{2\sqrt{c}}\right)^{d-1}$$

$$= \frac{\mathbb{1}_{\mathbb{R}_+}(r)}{Z_r^{\mathrm{R}}} e^{-\frac{r^2}{2\sigma^2}}\frac{1}{(2\sqrt{c})^{d-1}}\sum_{k=0}^{d-1}\binom{d-1}{k}\left(e^{\sqrt{c}r}\right)^{d-1-k}\left(-e^{-\sqrt{c}r}\right)^k$$

$$= \frac{\mathbb{1}_{\mathbb{R}_+}(r)}{Z_r^{\mathrm{R}}}\frac{1}{(2\sqrt{c})^{d-1}} e^{-\frac{r^2}{2\sigma^2}}\sum_{k=0}^{d-1}(-1)^k\binom{d-1}{k}e^{(d-1-2k)\sqrt{c}r}$$

$$= \frac{\mathbb{1}_{\mathbb{R}_+}(r)}{Z_r^{\mathrm{R}}}\frac{1}{(2\sqrt{c})^{d-1}}\sum_{k=0}^{d-1}(-1)^k\binom{d-1}{k}e^{-\frac{r^2}{2\sigma^2}+(d-1-2k)\sqrt{c}r}$$

$$= \frac{\mathbb{1}_{\mathbb{R}_+}(r)}{Z_r^{\mathrm{R}}}\frac{1}{(2\sqrt{c})^{d-1}}\sum_{k=0}^{d-1}(-1)^k\binom{d-1}{k}e^{\frac{(d-1-2k)^2}{2}c\sigma^2}e^{-\frac{1}{2\sigma^2}\left[r-(d-1-2k)\sqrt{c}\sigma^2\right]^2}. \tag{22}$$

### B.4.2 Sampling

In this section we detail the sampling scheme that we use for the Riemannian normal distribution $\mathcal{N}^{\mathrm{R}}_{\mathbb{B}^d_c}(\cdot|\boldsymbol{\mu},\sigma^2)$, along with a reparametrisation which allows to compute gradients with respect to the parameters $\boldsymbol{\mu}$ and $\sigma$.

**Sampling challenges due to the hyperbolic geometry**  Several properties of the Euclidean space do not generalise to the hyperbolic setting, unfortunately hardening the task of obtaining samples from *Riemannian* normal distributions. First, one can factorise a normal density through the space's dimensions – thanks to to the Pythagorean theorem – hence allowing to divide the task on several subspaces and then concatenate the samples. Such a property does not extend to the hyperbolic geometry, thus seemingly preventing us from focusing on 2-dimensional samples. Second, in Euclidean geometry, the polar radius $r$ is distributed according to $\rho^{\mathrm{W}}(r) = \frac{\mathbb{1}_{\mathbb{R}_+}(r)}{Z_r}e^{-\frac{r^2}{2\sigma^2}}r^{d-1}$, making it easy by a linear change of variable to take into account different scaling values. The non-linearity of $\sinh$ prevent us from using such a simple change of variable.

**Computing gradients with respect to parameters**  So as to compute gradients of samples $\boldsymbol{z}$ with respect to the parameters $\boldsymbol{\mu}$ and $\sigma$ of samples of a hyperbolic distributions, we respectively rely on the reparametrisation given by Eq 20 for $\nabla_{\boldsymbol{\mu}}\boldsymbol{z}$, and on an implicit reparametrisation (Figurnov et al., 2018) of $r$ for $\nabla_\sigma\boldsymbol{z}$. We have $\boldsymbol{z} = \exp^c_{\boldsymbol{\mu}}\left(\frac{r}{\lambda^c_{\boldsymbol{\mu}}}\boldsymbol{\alpha}\right)$ with $\boldsymbol{\alpha}\sim\mathcal{U}(\mathbb{S}^{d-1})$ and $r\sim\rho^{\mathrm{R}}(\cdot)$. Hence,

$$\nabla_{\boldsymbol{\mu}}\boldsymbol{z} = \nabla_{\boldsymbol{\mu}}\exp^c_{\boldsymbol{\mu}}(\boldsymbol{u}), \tag{23}$$

with $\boldsymbol{u} = \frac{r}{\lambda^c_{\boldsymbol{\mu}}}\boldsymbol{\alpha}$ (actually) independent of $\boldsymbol{\mu}$, and

$$\nabla_\sigma\boldsymbol{z} = \nabla_\sigma\exp^c_{\boldsymbol{\mu}}(\boldsymbol{u}) = \nabla_{\boldsymbol{u}}\exp^c_{\boldsymbol{\mu}}(\boldsymbol{u})\frac{\boldsymbol{\alpha}}{\lambda^c_{\boldsymbol{\mu}}}\nabla_\sigma r, \tag{24}$$

with $\nabla_\sigma(r)$ computed via the implicit reparametrisation given by

$$\begin{aligned}
\nabla_\sigma(r) &= -\left(\nabla_r F^{\mathrm{R}}(r,\sigma)\right)^{-1}\nabla_\sigma F^{\mathrm{R}}(r,\sigma) \\
&= -\left(\rho^{\mathrm{R}}(r;\sigma)\right)^{-1}\nabla_\sigma F^{\mathrm{R}}(r,\sigma).
\end{aligned} \tag{25}$$

**Sampling hyperbolic radii**  Unfortunately the density of the hyperbolic radius $\rho^{\mathrm{R}}(r)$ is not a well-known distribution and its cumulative density function does not seem analytically invertible. We therefore rely on rejection sampling methods.

**Adaptive Rejection Sampling**  By making use of the log-concavity of $\rho^{\mathrm{R}}$, we can rely on a piecewise exponential distribution proposal from adaptive rejection sampling (ARS) (R. Gilks and Wild, 1992). Such a proposal automatically adapt itself with respect to the parameters $\sigma$, $c$ and $d$. Even though $\mathcal{N}_{\mathbb{B}^d_c}$ is defined on a d-dimensional manifold, $\rho^{\mathrm{R}}$ is a univariate distribution hence the sampling scheme is not directly affected by dimensionality. The difficulty in ARS is to choose the initial set of points to construct the piecewise exponential proposal. To do so, we first compute the mean $m = \mathbb{E}_{r\sim\rho^{\mathrm{R}}}[r]$ and standard deviation $s = \mathbb{V}_{r\sim\rho^{\mathrm{R}}}[r]^{1/2}$ of the targeted distribution. Then we choose a grid $\eta = (\eta_1,\ldots,\eta_K) = (\mathrm{linspace}(\eta_{max},\eta_{min},K/2),\ \mathrm{linspace}(\eta_{min},\eta_{max},K/2))$. Eventually, we set the initial points $(x_1,\ldots,x_K)$ to $x_k = m + \eta_k*\min(s, 0.95*m/\eta_{max})$. For our experiments we chose $\eta_{min} = .1, \eta_{max} = 3, K = 20$. We do not adapt the proposal within the rejection sampling since we empirically found it unnecessary.

Alternatively, we derived bellow two non-adaptive proposal distributions along with their rejection rate constants. Yet, we observe that these rates do not scale well the dimensionality $d$ and distortion $\sigma$, making them ill-suited for practical purposes.

**Rejection Sampling with truncated Normal proposal**  The developed expression of $\rho(r)^{\mathrm{R}}$ from Eq (22) highlights the fact that the density can immediately be upper bounded by a truncated normal

density:

$$\rho(r)^{\mathrm{R}} = \frac{\mathbb{1}_{\mathbb{R}_+}(r)}{Z_r^{\mathrm{R}}} \frac{1}{(2\sqrt{c})^{d-1}} \sum_{k=0}^{d-1} (-1)^k \binom{d-1}{k} e^{\frac{(d-1-2k)^2}{2}c\sigma^2} e^{-\frac{1}{2\sigma^2}\left[r-(d-1-2k)\sqrt{c}\sigma^2\right]^2}$$

$$\leq \frac{\mathbb{1}_{\mathbb{R}_+}(r)}{Z_r^{\mathrm{R}}} \frac{1}{(2\sqrt{c})^{d-1}} \sum_{2k=0}^{d-1} \binom{d-1}{2k} e^{\frac{(d-1-4k)^2}{2}c\sigma^2} e^{-\frac{1}{2\sigma^2}\left[r-(d-1-4k)\sqrt{c}\sigma^2\right]^2}.$$

Then we choose our proposal $g$ to be the truncated normal distribution associated with $k = 0$, i.e. with mean $(d-1)\sqrt{c}\sigma^2$ and variance $\sigma^2$

$$g(r) = \frac{\mathbb{1}_{r>0}}{\sigma\left(1 - \Phi\left(-\frac{(d-1)\sqrt{c}\sigma^2}{\sigma}\right)\right)} \frac{1}{\sqrt{2\pi}} e^{-\frac{1}{2\sigma^2}\left(r-(d-1)\sqrt{c}\sigma^2\right)^2}$$

$$= \frac{1}{\sqrt{2\pi}} \frac{\mathbb{1}_{r>0}}{\sigma\left(1 - \frac{1}{2} - \frac{1}{2}\mathrm{erf}\left(-(d-1)\sqrt{c}\frac{\sigma}{\sqrt{2}}\right)\right)} e^{-\frac{1}{2\sigma^2}\left(r-(d-1)\sqrt{c}\sigma^2\right)^2}$$

$$= \sqrt{\frac{2}{\pi}} \frac{\mathbb{1}_{r>0}}{\sigma\left(1 + \mathrm{erf}\left(\frac{(d-1)\sqrt{c}\sigma}{\sqrt{2}}\right)\right)} e^{-\frac{1}{2\sigma^2}\left(r-(d-1)\sqrt{c}\sigma^2\right)^2}$$

$$= \frac{\mathbb{1}_{r>0}}{Z_g(\sigma)} e^{-\frac{1}{2\sigma^2}\left(r-(d-1)\sqrt{c}\sigma^2\right)^2} \tag{26}$$

with

$$Z_g = \sqrt{\frac{\pi}{2}}\sigma\left(1 + \mathrm{erf}\left(\frac{(d-1)\sqrt{c}\sigma}{\sqrt{2}}\right)\right). \tag{27}$$

Computing the ratio of the densities yield

$$\frac{\rho(r)^{\mathrm{R}}}{g(r)} = \frac{Z_g(\sigma)}{Z_r^{\mathrm{R}}} \frac{1}{(2\sqrt{c})^{d-1}} \sum_{k=0}^{d-1} (-1)^k \binom{d-1}{k} e^{\frac{(d-1-2k)^2}{2}c\sigma^2} e^{-\frac{1}{2\sigma^2}\left[r-(d-1-2k)\sqrt{c}\sigma^2\right]^2} e^{+\frac{1}{2\sigma^2}\left[r-(d-1)\sqrt{c}\sigma^2\right]^2}$$

$$= \frac{Z_g(\sigma)}{Z_r^{\mathrm{R}}} \frac{1}{(2\sqrt{c})^{d-1}} \sum_{k=0}^{d-1} (-1)^k \binom{d-1}{k} e^{\frac{(d-1-2k)^2}{2}c\sigma^2} e^{2k\sqrt{c}\left((d-1-k)\sqrt{c}\sigma^2-r\right)}.$$

Hence

$$\rho(r)^{\mathrm{R}}/g(r) \leq M \triangleq \frac{Z_g(\sigma)}{Z_r^{\mathrm{R}}} \frac{1}{(2\sqrt{c})^{d-1}} e^{\frac{(d-1)^2c\sigma^2}{2}}. \tag{28}$$

**Rejection Sampling with Gamma proposal** Now let's consider the following Gamma$(2, \sigma)$ density:

$$g(r) = \frac{\mathbb{1}_{r>0}}{Z_g(\sigma)} r e^{-\frac{r}{\sigma}}$$

with

$$Z_g(\sigma) = \Gamma(2)\sigma^2.$$

Then log ratio of the densities can be upper bounded as following:

$$\ln\left(\frac{\rho(r)^{\mathrm{R}}}{g(r)}\right) = \ln\frac{Z_g(\sigma)}{Z_r^{\mathrm{R}}} - \frac{r^2}{2\sigma^2} + (d-1)\ln(e^{\sqrt{c}r} - e^{-\sqrt{c}r}) - (d-1)\ln 2 - \ln r + \frac{r}{\sigma}$$

$$= \ln\frac{Z_g(\sigma)}{Z_r^{\mathrm{R}}} - (d-1)\ln 2 \underbrace{- \frac{r^2}{2\sigma^2} + \left((d-1)\sqrt{c} + \frac{1}{\sigma}\right)r}_{\leq \frac{((d-1)\sqrt{c}\sigma+1)^2}{2}} + \underbrace{(d-1)\ln\left(\frac{1 - e^{-2\sqrt{c}r}}{r}\right)}_{\leq(d-1)\ln(2\sqrt{c})}$$

$$\leq \ln\frac{Z_g(\sigma)}{Z_r^{\mathrm{R}}} + \frac{((d-1)\sqrt{c}\sigma + 1)^2}{2} + (d-1)\ln\sqrt{c}.$$

Hence

$$\rho(r)^{\mathrm{R}}/g(r) \leq M \triangleq \frac{Z_g(\sigma)}{Z_r^{\mathrm{R}}} c^{\frac{d-1}{2}} e^{\frac{((d-1)\sqrt{c}\sigma+1)^2}{2}}. \tag{29}$$

### B.4.3 Normalisation constant

In order to evaluate the density of the *Riemannian* normal distribution, we need to compute the normalisation constant, which we derive in this subsection.

**Cumulative density function**    First let's derive the cumulative density function of the hyperbolic radius. Integrating the expended density of Eq (22) yields

$$
\begin{aligned}
F_r^{\mathrm{R}}(r) &= \int_{-\infty}^{r} \rho^{\mathrm{R}}(r) dr \\
&= \frac{1}{Z_r^{\mathrm{R}}} \frac{1}{(2\sqrt{c})^{d-1}} \sum_{k=0}^{d-1} (-1)^k \binom{d-1}{k} e^{\frac{(d-1-2k)^2}{2} c\sigma^2} \\
&\quad \times \int_0^r e^{-\frac{1}{2\sigma^2}\left[r-(d-1-2k)\sqrt{c}\sigma^2\right]^2} dr \sum_{k=0}^{d-1} (-1)^k \binom{d-1}{k} e^{\frac{(d-1-2k)^2}{2} c\sigma^2} \\
&\quad \times \left[ \mathrm{erf}\left(\frac{r-(d-1-2k)\sqrt{c}\sigma^2}{\sqrt{2}\sigma}\right) \; \mathrm{erf}\left(\frac{(d-1-2k)\sqrt{c}\sigma}{\sqrt{2}}\right) \right]
\end{aligned}
\tag{30}
$$

with $\Phi : x \mapsto \frac{1}{2}\left(1 + \mathrm{erf}\left(\frac{x}{\sqrt{2}}\right)\right)$, the cumulative distribution function of a standard normal distribution.

**Taking the limit**    $F_r^{\mathrm{R}}(r) \xrightarrow[r\to\infty]{} 1$ in Eq (30) yield

$$
Z_r^{\mathrm{R}} = \sqrt{\frac{\pi}{2}} \sigma \frac{1}{(2\sqrt{c})^{d-1}} \sum_{k=0}^{d-1} (-1)^k \binom{d-1}{k} e^{\frac{(d-1-2k)^2}{2} c\sigma^2} \left[1 + \mathrm{erf}\left(\frac{(d-1-2k)\sqrt{c}\sigma}{\sqrt{2}}\right)\right].
\tag{31}
$$

Note that by the antisymmetry of erf, one can simplify Eq (31) with a sum over $\lceil d/2 \rceil$ terms (as done in Hauberg (2018)). Also, computing such a sum is much more stable by relying on the *log sum exp* trick. Integrating Equation (19) of Appendix B.2 gives

$$
Z^{\mathrm{R}} = Z_r^{\mathrm{R}} Z_{\boldsymbol{\alpha}}
\tag{32}
$$

As a reminder, the surface area of the $d-1$-dimensional hypersphere with radius 1 is given by

$$
Z_\alpha = A_{\mathbb{S}^{d-1}} = \frac{2\pi^{d/2}}{\Gamma(d/2)}.
$$

For the special case of $c = 1$ and $d = 2$ we recover the formula given in Said et al. (2014)

$$
Z_r^{\mathrm{R}} = \sqrt{\frac{\pi}{2}} \sigma e^{\frac{\sigma^2}{2}} \mathrm{erf}\left(\frac{\sigma}{\sqrt{2}}\right).
$$

### B.4.4 Expectation of hyperbolic radii

Computing the expectation of the hyperbolic radius $r \sim \rho^{\mathrm{R}}$ is of use to choose the initial set of points to construct the piecewise exponential proposal. By integrating the expended density of Eq (22), we

get

$$\mathbb{E}[r] = \int_{-\infty}^{\infty} r\rho^{\mathrm{R}}(r)dr$$

$$= \frac{1}{Z_r^{\mathrm{R}}} \frac{1}{(2\sqrt{c})^{d-1}} \sum_{k=0}^{d-1} (-1)^k \binom{d-1}{k} e^{\frac{(d-1-2k)^2}{2}c\sigma^2} \int_0^{\infty} re^{-\frac{1}{2\sigma^2}\left[r-(d-1-2k)\sqrt{c}\sigma^2\right]^2} dr$$

$$= \frac{1}{Z_r^{\mathrm{R}}} \frac{1}{(2\sqrt{c})^{d-1}} \sum_{k=0}^{d-1} (-1)^k \binom{d-1}{k} e^{\frac{(d-1-2k)^2}{2}c\sigma^2}$$

$$\times \left[ \sqrt{\frac{\pi}{2}} (d-1-2k)\sqrt{c}\sigma^2\sigma \left(1 + \mathrm{erf}\left(\frac{(d-1-2k)\sqrt{c}\sigma}{\sqrt{2}}\right)\right) + \sigma^2 e^{-\frac{(d-1-2k)^2 c\sigma^2}{2}} \right]$$

$$= \frac{1}{Z_r^{\mathrm{R}}} \sqrt{\frac{\pi}{2}}\sigma \frac{1}{(2\sqrt{c})^{d-1}} \sum_{k=0}^{d-1} (-1)^k \binom{d-1}{k}$$

$$\times \left[ e^{\frac{(d-1-2k)^2}{2}c\sigma^2} (d-1-2k)\sqrt{c}\sigma^2 \left(1 + \mathrm{erf}\left(\frac{(d-1-2k)\sqrt{c}\sigma}{\sqrt{2}}\right)\right) + \sigma\sqrt{\frac{2}{\pi}} \right]$$

$$\tag{33}$$

## C  Experimental details

In this section we give more details on the datasets, architecture designs and optimisation schemes used for the experimental results given in Section 5.

### C.1  Synthetic Branching Diffusion Process

**Generation**  Nodes $(\boldsymbol{y}_1, \ldots, \boldsymbol{y}_N) \in \mathbb{R}^n$ of the branching diffusion process are sampled as follow

$$\boldsymbol{y}_i \sim \mathcal{N}\left(\cdot \,|\, \boldsymbol{y}_{\pi(i)}, \sigma_0^2\right) \quad \forall i \in 1, \ldots, N$$

with $\pi(i)$ being the index of the $i$th node's ancestor and $d(i)$ its depth. Then, noisy observations are sampled for each node $\boldsymbol{x}_i$,

$$\boldsymbol{x}_{i,j} = \boldsymbol{y}_i + \boldsymbol{\epsilon}_{i,j}, \quad \boldsymbol{\epsilon}_{i,j} \sim \mathcal{N}\left(\cdot \,|\, \boldsymbol{0}, \sigma_j^2\right) \quad \forall i, j.$$

The root $x_0$ is set to $\boldsymbol{0}$ for simplicity. The observation dimension is set to $n = 50$. The dataset $(\boldsymbol{x}_{i,j})_{i,j}$ is centered and normalised to have unit variance. Thus, the choice of variance $\sigma_0^2$ does not matter and it is set to $\sigma_0 = 1$. The number of noisy observations is set to $J = 5$, and its variance to $\sigma_j^2 = \sigma_0^2/5 = 1/5$. The depth is set to 6 and the branching factor to 2.

**Architectures**  Both $\mathcal{N}$-VAE and $\mathcal{P}^c$-VAE decoders parametrise the mean of the unit variance Gaussian likelihood $\mathcal{N}(\cdot|f_{\boldsymbol{\theta}}(\boldsymbol{z}), 1)$. Their encoders parametrise the mean and the log-variance of respectively an isotropic normal distribution $\mathcal{N}(\cdot|g_{\boldsymbol{\phi}}(\boldsymbol{z}))$ and an isotropic hyperbolic normal distribution $\mathcal{N}_{\mathbb{B}_c^d}(\cdot|g_{\boldsymbol{\phi}}(\boldsymbol{z}))$. The $\mathcal{N}$-VAE's encoder and decoder are composed of 2 Fully-Connected layers with a ReLU activation in between, as summed up in Tables 5 and 6. The $\mathcal{P}^c$-VAE's design is similar, the differences being that the decoder's output is mapped to manifold via the exponential map $\exp_{\boldsymbol{0}}^c$, and the decoder's first layer is made of *gyroplane units* presented in Section 3.2, as summarised in Tables 7 and 8. Observations live in $\mathcal{X} = \mathbb{R}^{50}$ and the latent space dimensionality $d$ is set to $d = 2$.

| Layer | Output dim | Activation |
|-------|-----------|------------|
| Input | 50 | Identity |
| FC | 200 | ReLU |
| FC | 2, 1 | Identity |

Table 5: Encoder network for $\mathcal{N}$-VAE

| Layer | Output dim | Activation |
|-------|-----------|------------|
| Input | 2 | Identity |
| FC | 200 | ReLU |
| FC | 50 | Identity |

Table 6: Decoder network for $\mathcal{N}$-VAE

Table 7: Encoder network for $\mathcal{P}^c$-VAE

| Layer | Output dim | Activation |
|-------|-----------|------------|
| Input | 50 | Identity |
| FC | 200 | ReLU |
| FC | 2, 1 | $\exp_\mathbf{0}^c$, Identity |

Table 8: Decoder network for $\mathcal{P}^c$-VAE

| Layer | Output dim | Activation |
|-------|-----------|------------|
| Input | 2 | Identity |
| Gyroplane | 200 | ReLU |
| FC | 50 | Identity |

The synthetic datasets are generated as described in Section 5, then centred and normalised to unit variance. There are then randomly split into training and testing datasets with a proportion $0.7$.

**Optimisation**   Gyroplane offset $\boldsymbol{p} \in \mathbb{B}_c^d$ are only *implicitly* parametrised to live in the manifold, by projecting a real vector $\boldsymbol{p} = \exp_\mathbf{0}^c(\boldsymbol{p}')$. Hence, all parameters $\{\boldsymbol{\theta}, \boldsymbol{\phi}\}$ of the model explicitly live in Euclidean spaces which means that usual optimisation schemes can be applied. We therefore rely on Adam optimiser (Kingma and Ba, 2016) with parameters $\beta_1 = 0.9$, $\beta_2 = 0.999$ and a constant learning rate set to $1e - 3$. Models are trained with mini-batches of size 64 for 1000 epochs. The ELBO is approximated with a MC estimate with $K = 1$.

**Baselines**   The principal component analysis (PCA) embeddings are obtained via a singular-value decomposition (SVD) by projecting the dataset on the basis associated with the two highest singular values. The Gaussian process latent variable model (GPLVM) embeddings are obtained by maximising the marginal likelihood of a (non-Bayesian) GPLVM with RBF kernel, and whose latent variables are initialised with PCA.

## C.2   MNIST digits

The MNIST dataset (LeCun and Cortes, 2010) contains 60,000 training and 10,000 test images of ten handwritten digits (zero to nine), with 28x28 pixels.

**Architectures**   The architectures used for the encoder and the decoder for Mnist are similar to the ones used for the Synthetic Branching Diffusion Process. They differ by the dimensions of the observation space ($\mathcal{X} = \mathbb{R}^{28 \times 28}$) and hidden space. The output of the first fully connected layer is here equal to 600. The latent space dimensionality $d$ is set to 2, 5, 10 and 20 respectively. The bias of the decoder's last layer is set to the average value of digits (for each pixel). The architectures used for the classifier are similar than the decoder architectures, the only difference being the output dimensionality (10 labels). We initialise the classifier's first layer with decoder's first layer weights. Then the classifier is trained to minimise the cross entropy for 5 epochs, with mini-batches of size 64 and a constant learning rate of $1e-3$.

**Optimisation**   We use $[0, 1]$ normalised data as targets for the mean of a Bernoulli distribution, using negative cross-entropy for $\log p(\boldsymbol{x}|\boldsymbol{z})$. We set the prior distribution's distortion to $\sigma = 1$. We rely on Adam optimiser with parameters $\beta_1 = 0.9$, $\beta_2 = 0.999$ and a constant learning rate of $5e^{-4}$. Models are trained with mini-batches of size 128 for 80 epochs.

## C.3   Graph embeddings

The PhD advisor-advisee relationships graph (Nooy et al., 2011) contains 344 nodes and 343 edges. The phylogenetic tree expressing genetic heritage (Hofbauer et al., 2016; Sanderson and Eriksson, 1994) contains 1025 nodes and 1043 edges. The biological set representing disease relationships (Goh et al., 2007; Rossi and Ahmed, 2015) contains 516 nodes and 1188 edges. We follow the training and evaluation procedure introduced in Kipf and Welling (2016).

**Architectures**   We also follow the featureless architecture introduced in Kipf and Welling (2016), namely a two-layer GCN with 32 hidden dimensions to parametrise the variational posteriors, and a likelihood which factorises along edges $p(\boldsymbol{A}|\boldsymbol{Z}) = \prod_{i=1}^{N} \prod_{j=1}^{N} p(A_{ij}|\boldsymbol{z}_i, \boldsymbol{z}_j)$, with $\boldsymbol{A}$ being the adjacency matrix. The probability of an edge is defined through the latent metric by $p(A_{ij} = 1|\boldsymbol{z}_i, \boldsymbol{z}_j) = 1 - \tanh(d_\mathcal{M}(\boldsymbol{z}_i, \boldsymbol{z}_j))$. For the Poincaré ball latent space, the encoder output is projected

on the manifold: $\boldsymbol{\mu} = \exp_{\mathbf{0}}(\mathrm{GCN}_{\boldsymbol{\mu}}(\boldsymbol{A}))$. The latent dimension is set to 5 for the experiments. We use a *Wrapped* Gaussian prior and variational posterior.

**Optimisation**  We use the adjacency matrix $\boldsymbol{A}$ as target for the mean of a Bernoulli distribution, using negative cross-entropy for $\log p(\boldsymbol{A}|\boldsymbol{Z})$. We rely on Adam optimiser with parameters $\beta_1 = 0.9$, $\beta_2 = 0.999$ and a constant learning rate of $1e^{-2}$. We perform full-batch gradient descent for 800 epochs and make use of the reparametrisation trick for training.

# D  More experimental qualitative results

Figure 9 shows latent representations of $\mathcal{P}^c$-VAEs with different curvatures. With "small" curvatures, we observe that embeddings lie close the center of the ball, where the geometry is close to be Euclidean. Similarly as Figure 9, Figure 10 illustrates the learned latent representations of $\mathcal{P}^c$-VAE with decreasing curvatures $c$, by highlighting the leaned *gyroplanes* of the decoder.

Figure 9: Branching diffusion process latent representations of $\mathcal{P}^c$-VAE with decreasing curvatures $c = 1.2, 0.3, 0.1$ (Left to Right).

Figure 10: Branching diffusion process latent representations of $\mathcal{P}^c$-VAE with decreasing curvatures $c = 1.2, 0.3, 0.1$ (Left to Right).

Figure 11: Branching diffusion process latent representation of $\mathcal{P}^1$-VAE (Left) and $\mathcal{N}$-VAE (Right) with heatmap of the log distance to the hyperplane (in pink).

Figure 12: MNIST average confusion matrices of the classifiers trained on embeddings from the $\mathcal{P}^{1.4}$-VAE (Left) and $\mathcal{N}$-VAE (Right) models.