[Reviews · NeurIPS 2019]

Reviewer 1



To the best of my knowledge the idea of using either the Riemannian normal or the wrapped normal in a VAE is new and worth of attention. The paper is excellently written and very clear. 1) Line 48: "We show that a VAE endowed with a hyperbolic latent space will be more capable at representing and discovering hierarchies than traditional VAEs that use a Euclidean latent space". While I do understand what you mean I feel like this line is a bit misleading. It is true that your VAE samples from a hyperbolic space and that a standard VAE samples in a euclidean space but there is no guarantee that the resulting latent spaces are respectively hyperbolic or euclidean. Would it be possible to change this sentence? 2) In table 1 you measure the negative test marginal likehood with L_IWAE. But you never define what L_IWAE is. Could you define it properly? 3) Could you move figure 5? It is refereed a whole page after been introduced and it is a bit confusing. You cite figure 6 before 5 in the text. 4) Line 183: "Derived earlier in Eq 8 and 11". The derivation is in the appendix, could you change this line to make it more clear? It is a bit confusing right now. To summarise, this is an excellent paper that introduces a theoretically sound modification of the standard VAE and achieves state of the art results on a series of baselines. -------------------------------------------------------------- Post rebuttal: The authors agreed about fixing my main complain therefore I will maintain my overall score and argue for acceptance.

Reviewer 2



--- Review update: The authors clarified the details of the ablation study in Figure 5, so now I am convinced that the proposed updates to the decoder architecture constitute a significant improvement. Therefore, I am increasing my score to 7. --- The authors consider variational autoencoders with hidden variable $z$ in hyperbolic space. Intuitively, hyperbolic space is suitable for learning hierarchical representations. The exponential growth of surface with radius allows locating an exponential number of leaves of a tree on an equal distance. In the concurrent work, Nagano2018 considered analogous models. The submission has two principal differences. First, it studies a different type of distribution on a hyperbolic plane as a building block of VAE. Second, it uses a particular “hyperbolic” layer in the decoder. The former leads to better results for one of the tasks in the experimental section, but the paper does not study the effect of the layer. In general, the paper is well-written and technically sound. The claims are supported by thoroughly described experimental results. However, the use of hyperbolic spaces is motivated by two-dimensional illustration. How does the analogy between trees and hyperbolic spaces work beyond two-dimensional case?

Reviewer 3



-Originality: Doing variational inference based on a standard ELBO with reparamterised gradients on the Poincare ball is new. It uses ideas similar to very recent/concurrent work (Ganea et al., 2018; Ovinnikov, 2018; Nagano et al., 2019), but it is made clear how this work differs from related work. Quality: The submission seems technically sound, with detailed experimental results. The paper empirically compares their approach mostly with their Euclidean counterpart. This is fair, of course, but it would be interesting to see how it compares empirically with the Poincaré Wasserstein Autoencoder (Ovinnikov, 2019) and the hyperboloid model of Nagano et al. (2019), like do they yield similar latent representations, how are the respective sample qualities? -Clarity: The paper is polished and well written. The background on Riemannian geometry is to the point, so that the paper is in most parts accessible to readers without training in non-Euclidean geometry. Nevertheless, I feel that readers could benefit from more high-level guidance in Appendix B, like what do we learn from Section B.8 and B.9? -Significance: I feel that this is a significant work and others can build on these ideas either methodologically or experimentally. The experiments presented show advantages over Euclidean counterparts in different domains. I also feel that the proposed approach is easier to use for practitioners compared to some related work. -Some comments/thoughts: I was wondering if instead of using a Gaussian, similar extensions can be made for other spherical distributions such as a Student-t, as they also have a stochastic representation of z=r*\alpha (Fang et al, Symmetric Multivariate and Related Distributions, 1990, Chapter 2)? Also Mallasto et al, Probabilistic Riemannian submanifold learning with wrapped Gaussian process latent variable models, 2019, considered the pushforward of a Gaussian Process by the exponential map. Could something like this be used here for a VAE on the Poincare ball, say with a GP prior (Casale et al, 2018)? Further, the Riemannian Normal distribution seems to be unstable in higher dimensions from Table 3. Are there different limiting distributions of the Wrapped/Riemannian model as the latent state space dimension goes to infinity, and does this provide some guidance for applications? POST AUTHOR RESPONSE: Having read the rebuttal and the other reviews, I keep my initial score of 7. The authors agreed to improve Appendix B, which I felt lacked some high-level guidance. Their response also indicated that the proposed approach can be extended in different ways. My main complaint was that the paper lacks some empirical comparison with very recent related work (Ovinnikov, 2019, Nagano et al., 2019). However, even without such a comparison, I think it is still a complete and interesting paper.

[Author Response · NeurIPS 2019]

We thank the reviewers for their time, helpful feedback, and advice. Overall, the reviewers praised the originality and
clarity of the work. We thank them for their kind words, and hope to address any remaining concerns below.

**(R1) Line 48 may be misleading.** We agree and propose the following replacement: "We show that replacing VAE
latent space components, which traditionally assume a Euclidean metric over the latent space, by their hyperbolic
generalisation helps to represent and discover hierarchies." In particular, the prior and posterior probability densities
are defined w.r.t. the volume induced by the metric tensor, the decoder (as opposed to concurrent work) treats latent
variables as points in the Poincaré ball (by computing geodesic distances to hyperplanes) and the encoder projects
points via the exponential map.

**(R2) asked for evidence that the "hyperbolic" decoder was helpful.** We would like to point out that we conducted
an ablation study on this point, whose results are summarised in Figure 5. Admittedly, the figure and the related
explanations are a bit condensed. We will improve that for the next version.

In more detail, we compared three decoders: (i) a standard "vanilla" multilayer perceptron (implicitly relying on the
flat Euclidean geometry), (ii) a MLP precomposed by the logarithm map defined at the centre of the ball (can be seen
as a linearisation of the manifold) and (iii) a decoder with a "hyperbolic" layer – described in Section 2.3 – which
generalises a linear layer by computing geodesic distances to hyperplanes. Figure 5 shows improvements in terms of
marginal log-likelihood estimates relative to the MLP baseline (i) for different latent dimensions and computed on the
MNIST dataset. This ablation study shows that linearising the Poincaré ball through the logarithm map (i.e. decoder
(ii)) before feeding latent variables through an MLP improves the performance compared to a vanilla MLP decoder. Yet,
the composition of the logarithm map and a linear layer (i.e. decoder (i)) is not as good as a "hyperbolic" layer (i.e.
decoder (iii)) which directly rely on the geometry of the Poincaré ball. Though, the differences in performances shrinks
when the latent dimension increases.

For a detailed explanation of the analogy between trees and the hyperbolic space, we recommend reading section II of
Krioukov et al. (2010). The analogy is not limited to the two-dimensional case. Although, as shown in De Sa et al.
(2018), a two dimensional hyperbolic space is sufficient to embedded trees with arbitrarily low reconstruction error, if
one has access to an arbitrarily high number of bits of precision.

**(R4) Empirical comparison to related methods.** We agree that comparing our method against concurrent work is
indeed important. Unfortunately, the code or necessary experimental details have not been released, preventing us from
doing so.

**(R1, R4) Clarifications and minor typos.** Thank you for pointing out some minor typos and places where clarifica-
tions could be useful. We will correct the typos in the text and move Figure 5 to the next page.

To clarify for (R1), $\mathcal{L}_{IWAE}$ refers to the IWAE unbiased estimate of the marginal likelihood introduced in Burda et al.
(2015). We used 5000 samples in our experiments. We will include this definition in the next draft, thank you for
pointing out our omission.

(R4) asked for additional high-level guidance in Appendix B. Thank you for the suggestion, we will reorder the
subsections and write better connections between them so as to ease the reading.

**(R4) Possible generalisations.** Thank you for your suggestions. Indeed, we believe spherical distributions can be
extended in a similar fashion. One could consider a wrapped Student-t as $Z \sim \exp_{\mu\#} S_t(0, \nu)$, or a Riemannian
Student-t with density (w.r.t. to the measure induced by the metric tensor) proportional to $\left(1 + d_M(z, \mu)^2/\nu\right)^{(-\nu+1)/2}$.
As you point out, one could put a wrapped Gaussian process prior on the Poincaré ball to break the independence
assumption between latent variables in VAEs. Concerning the limiting behaviour of the hyperbolic normal distributions,
it appears that they are different, as the dimension space goes to infinity. Though in the context of dimensionality
reduction, we believe that hyperbolic spaces are mostly useful in the low-dimensional setting.

## References

Burda, Y., Grosse, R. B., and Salakhutdinov, R. (2015). Importance weighted autoencoders. *CoRR*, abs/1509.00519.

De Sa, C., Gu, A., Re, C., and Sala, F. (2018). Representation Tradeoffs for Hyperbolic Embeddings. *arXiv.org*.

Krioukov, D., Papadopoulos, F., Kitsak, M., Vahdat, A., and Boguna, M. (2010). Hyperbolic Geometry of Complex
Networks. *arXiv.org*, (3):253.


[Meta-Review · NeurIPS 2019]

This paper examines an alternative latent space, with sensible ablation studies, and sensible proposals for modifying the rest of the architecture to match. Our main complaint is that the paper lacks some empirical comparison with very recent related work (Ovinnikov, 2019, Nagano et al. , 2019). However, even without such a comparison, we think it is still a complete and interesting paper.